

# Evaluation of a new inference method for estimating ammonia volatilisation from multiple agronomic plots

Benjamin Loubet[1,*], Marco Carozzi[1#], Polina Voylokov[1], Jean-Pierre Cohan[2], Robert Trochard[2], Sophie Génermont[1]

1 INRA, UMR ECOSYS, INRA, AgroParisTech, Université Paris-Saclay, 78850, Thiverval-Grignon, France
2 ARVALIS, Institut du Végétal, 91720, Boigneville, France
# now at: Agroscope Research Station, Climate and Air Pollution Group, Zurich, Switzerland

* Corresponding author: Benjamin.Loubet@inra.fr

**Abstract.** Tropospheric ammonia ($NH_3$) is a threat to the environment and human health and is mainly emitted by agriculture. Ammonia volatilisation following application of nitrogen in the field accounts for more than 40% of the total ammonia emissions in France. This hence represents a major loss of nitrogen use efficiency which needs to be reduced by appropriate agricultural practices. In this study we evaluate a novel method to infer ammonia volatilisation from small agronomic plots made of multiple treatments with repetition. The method is based on the combination of a set of ammonia diffusion sensors exposed for durations of 3 hours to 1 week, and a short-range atmospheric dispersion model, used to retrieve the emissions from each plot. The method is evaluated by mimicking ammonia emissions from an ensemble of 9 plots with a resistance-analogue-compensation-point surface exchange scheme over a yearly meteorological database separated into 28-days periods. A multi-factorial simulation scheme is used to test the effects of sensor number and heights, plot dimensions, source strengths and background concentrations, on the quality of the inference method. We further demonstrate by theoretical considerations in the case of an isolated plot that inferring emissions with diffusion sensors integrating over daily periods will always lead to underestimations due to correlations between emissions and atmospheric transfer. We evaluated these underestimations as -8% ± 6% of the emissions for a typical western European climate. For multiple plots, we find that this method would lead to median underestimations of -16% with an interquartile [-8% -22%] for two treatments differing by a factor of up to 20 and a control treatment with no emissions. We further evaluate the methodology for varying background concentrations and ammonia emission patterns and demonstrate the low sensitivity of the method to these factors. The method was also tested in a real case and proved to provide sound evaluations of ammonia losses from surface applied and incorporated slurry. We hence showed that this novel method should be robust and suitable for estimating ammonia emissions from agronomic plots. Further work should anyway be produced for validating this method in real conditions.

**Keywords:** $NH_3$ emission, multiple sources, dispersion modelling, experimental design, diffusive samplers

## Introduction

Tropospheric ammonia ($NH_3$) is mainly emitted by agriculture and has great environmental impacts (atmospheric pollution, eutrophication, reduction of biodiversity) which are increasingly taken into account in European and international regulations (Council, 1996; Council, 2016; UNECE, 2012). Ammonia losses also have great agronomic and economic impacts for farmers, as it reduces nitrogen use efficiency. The varying



prices of mineral fertilizers and concerns about environmental and health threats demand improvements in the
efficiency of nitrogen utilisation, and especially in recycling nitrogen through organic fertilization (Sutton et al.,
2011). Indeed, $NH_3$ volatilization during storage of manure and slurry and following their field application is the
main source of $NH_3$ in Europe (55.3% of the emissions) and France (48-65%) while farm buildings emissions
represent 44.7% and 25-50% in Europe and France, respectively (CITEPA, 2017; ECETOC, 1994; EUROSTAT,
2012; Faburé et al., 2011). Reducing $NH_3$ losses from this agricultural sector is therefore a major objective for
applied research.
While $NH_3$ emissions from farm buildings and storage can be handled by engineering solutions, losses during
organic fertilisation are much more dependent on the combination of application methods (splash plate, band
spreading, pressurised injection, open and close slot injection, trailing hose and trailing shoe), soil type and
occupation, and environmental conditions (soil humidity, air temperature, wind speed, solar radiation) (Sommer
et al., 2003). For instance, Sintermann et al. (2012) report $NH_3$ losses following cattle and pig slurry application
in the field ranging from a few percent to 50% over large fields and up to 100% over medium fields. Evaluating
ammonia losses from field fertilisation over a range of practices, soil and climatic conditions is therefore key in
evaluating the best application methods.
However, characterising these emissions at the field scale requires complex experimental design and most of the
time large fields (Ferrara et al., 2016; Ferrara et al., 2012; Flechard and Fowler, 1998; Loubet et al., 2012;
Milford et al., 2009; Sintermann et al., 2011b; Spirig et al., 2010; Sun et al., 2015; Whitehead et al., 2008).
Especially useful for measuring ammonia losses are methods that can deal with small and medium-scale fields
like agronomic trials (squares of 20 to 50 m on the side), which are widespread. Indirect estimation methods (soil
nitrogen balance or $^{15}N$ balance) are not well adapted to evaluate gaseous ammonia losses, mainly because of the
soil heterogeneity and also because the method relies on evaluating small variations of large numbers (McGinn
and Janzen, 1998). Among existing methods for measuring $NH_3$ emissions, the integrated horizontal flux method
(Wilson and Shum, 1992) is well adapted, but is a subject of debate in its practical application since it seem to be
systematically biased towards higher estimates (Häni et al., 2016; Sintermann et al., 2012). Alternatively,
enclosure methods proved to be not representative for a sticky compound such as ammonia (Pacholski et al.,
2006), but more concerning is the fact that ammonia fluxes result from an air-surface equilibrium which is
disturbed by the confined environment offered by the chamber. Inverse dispersion modelling approaches either
based on backward Lagrangian Stochastic models (Flesch et al., 1995) or Eulerian models (Kormann and
Meixner, 2001; Loubet et al., 2001), based on the Philip equation (Philip (1959) have been demonstrated to be
adapted for estimating $NH_3$ volatilization from intensive sources (Loubet et al., 2010; Sommer et al., 2005).
These approaches are well adapted to small or medium fields ($\leq 50 \times 50$ m$^2$) but typically require hourly
concentrations. Long term concentration measurements of $NH_3$ are now well handled by the use of short path
passive samplers developed by Sutton, et al. (2001), or active denuders, which have both been used for
concentration monitoring for years (Tang et al., 2001; Tang et al., 2009). These active denuders can be adapted
for measuring fluxes based on conditional sampling like the conditional time averaged gradient method COTAG
(Famulari et al., 2010), which is a useful method but only adapted for large fields ($\geq 0.5$ ha). The passive
samplers have also been shown to be adapted for inverse modelling estimations of $NH_3$ sources for large fields
(Carozzi et al., 2013b; Ferrara et al., 2014).



In another field of research, solutions to the multiple source problem, which consists of inferring multiple sources based on measured concentrations at multiple points in space and time, have been developed especially since 2008 (Crenna et al., 2008; Gao et al., 2008; Gericke et al., 2011; Mukherjee et al., 2015; Vandré and Kaupenjohann, 1998). They have chiefly been used over regional scales (Flesch et al., 2009; Lushi and Stockie, 2010; Yee and Flesch, 2010), and have been shown to be very dependent on the source-sensor geometry (Crenna et al., 2008; Flesch et al., 2009; Wang et al., 2013 ). Mukherjee et al. (2015) highlighted the dependency of the inferred source to background concentration and plot disposition, by means of an inverse footprint approach. Yee et al. (2008) have shown how to retrieve the number, location and intensity of multiple sources with dispersion models coupled with Bayesian inference methods. Yee and Flesch (2010) have evaluated the inversion and inference methods for determining 4 points sources using several laser transects. Flesch et al. (2009) have shown that source-receptor geometry is critical in determining whether a multiple-source inversion problem can provide realistic solutions or not. Flesch et al. (2009) have moreover shown that if the geometry is well chosen the accuracy of the method for 15 min data can reach 10% to 20%. These studies have also shown that the multiple source inversion problems can be solved if not ill-conditioned (ill-conditioning depends on the location of sources and concentration sensors and is characterised by a conditioning number $\kappa$).

In this study, we pose the following research questions: **"Can inverse dispersion modelling approaches be used for inferring NH$_3$ emissions from multiple small plots (agronomic trials) using passive samplers, and to which degree of accuracy?"** The answer is given through the investigation of the optimal design in terms of field dimensions, plots locations and size, passive sampler locations and their duration of exposure. Throughout this study, agronomic trials are considered as adjacent multiple small fields with repetitions of *treatments*. A typical trial would consist of three repetitions of three treatments. Hence the double challenge that we face in this study is (i) to consider together the multiple source issue (adjacent small fields) and the (ii) time-integration issue (using passive samplers).

To answer these questions, we use a 4 step approach: (1) The ammonia emissions are first modelled on each source using prescribed NH$_3$ emission potential dynamics coupled with a simple soil-vegetation-atmosphere exchange scheme to mimic realistic seasonal, daily and hourly variations in NH$_3$ emissions. (2) These prescribed emissions are then used to estimate the concentration at each target location using short-range atmospheric dispersion modelling over half hourly periods. (3) The obtained concentrations are then averaged over several integration periods to simulate the behaviour of passive samplers. Finally, (4) the sources are evaluated by inference with dispersion modelling based on the averaged concentrations.

Two dispersion models and several inference methodologies are evaluated. The effect of the size of the source, the locations of targets, the dynamics and magnitude of each source and the meteorological conditions are evaluated and discussed. The feasibility of the method is finally evaluated over a real case with two repetitions of three treatments (slurry spreading, injection and a reference without fertilisation).

## 2. Materials and methods

At first we present the theoretical background of source inference by optimisation for single and multiple sources with time averaging concentration sensors. Then the method used to generate a realistic ammonia source is explained and the dispersion models used for generating the concentration fields and inferring back the sources





are presented. The geometry of the sources and sensors and the meteorological data used are then shown, and
finally the real test case used for evaluating the method is detailed.

### 2.1 The theory of the source inference method

At first we will recall some important theoretical features of the "inverse dispersion modelling" approach which
is actually an inference method.

#### 2.1.1 Case of a single area source and a single concentration sampler

We first consider the case of a single area source with a single concentration sampler (target). The source is
varying with time. The method is based upon the general superimposition principle (Thomson et al., 2007),
which relates the concentration at a given location $C(x,t)$ to the source strength $S(t)$ and the background
concentration $C_{bgd}(t)$ using a transfer function $D(x,t)$, which has the dimensions of a transfer resistance (s m$^{-1}$).
$$C(x,t) = D(x,t) \times S(t) + C_{bgd}(t) \qquad (1)$$
Here $x$ denotes the location of the sensor and $t$ the time. The superimposition principle implies that the studied
tracer must be conservative, which is a reasonable hypothesis for $NH_3$ whose reaction time with acids in the
atmosphere is below the transport time for spatial scales below 1000 m (Nemitz et al., 2009). Moreover, in **Eq.**
**(1)**, we assume a spatially homogeneous area source with strength $S(t)$. The spatial homogeneity of the source is
less trivial for $NH_3$ as the source itself depends on the concentration at the surface. However (Loubet et al.,
2010) have shown that the heterogeneity of the source can be neglected as long as the dimension of the source is
larger than 20 m. Hence, this study is limited to source areas with fetch larger than 20 m and a spread of the
concentration samplers over a domain smaller than 1000 m. Moreover, it is interesting to note that for infinitely
spread fields, the transfer resistance is linearly linked to the transfer matrix (See supplementary material S1)

#### 2.1.2 Effect of time averaging sensors on source inference for a single source

Since we consider time averaging concentration samplers, we develop the time-averaged equation of **Eq. (1)**
over a time period τ :
$$\overline{C(x)}^{\tau} = \overline{D(x) \times S}^{\tau} + \overline{C_{bgd}}^{\tau} \qquad (2)$$
where the overbars denote a time average over the period τ. Similarly as what is done in turbulent flux
calculations, the first part of the right hand side of **Eq. (2)** is decomposed using the Reynolds decomposition of a
random variable (Kaimal and Finnigan, 1994), giving:
$$\overline{C(x)}^{\tau} = \overline{D(x)}^{\tau} \times \overline{S}^{\tau} + \overline{C_{bgd}}^{\tau} + \overline{D'(x)S'}^{\tau} \qquad (3)$$

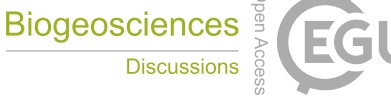

149 where $\overline{D(x)'S'}^{\tau}$ is the time covariance between $D(x,t)$ and $S(t)$. If the averaged background concentration $\overline{C_{bgd}}^{\tau}$

150 is a known quantity, **Eq. (3)** can be easily manipulated to give an estimation of the averaged source strength $\overline{S}^{\tau}$,

151 the quantity we want to infer:

152

153 $$\overline{S}^{\tau} = \frac{\overline{C(x)}^{\tau} - \overline{C_{bgd}}^{\tau}}{\overline{D(x)}^{\tau}} - \frac{\overline{D'(x)S'}^{\tau}}{\overline{D(x)}^{\tau}} \qquad (4)$$

154      (I)     (II)

155 In the right hand side of **Eq. (4)**, (I) can be calculated from measured $\overline{C_{bgd}}^{\tau}$ and $\overline{C(x)}^{\tau}$ and $\overline{D(x)}^{\tau}$ which is itself

156 calculated with dispersion models. On the contrary (II) is *a priori* unknown and depends on the correlation

157 between the source strength and the transfer function $\overline{D(x)'S'}^{\tau}$. Hence, if (II) is neglected, the inferred source $\overline{S}^{\tau}$

158 is biased. The relative bias of the method is then:

159

160 $$\frac{\delta \overline{S}^{\tau}}{\overline{S}^{\tau}} = \frac{\overline{D'(x)S'}^{\tau}}{\overline{D(x)}^{\tau} \times \overline{S}^{\tau}} \qquad (5)$$

161

162 Hence we show in **Eq. (5)** that time-averaging leads to a relative bias which can be quantified by the time

163 covariance between the transfer function and the source strength. However this quantity is by nature unknown

164 since the dynamics of $S(t)$ is unknown. Determining $\overline{D(x)'S'}^{\tau}$ requires knowledge of the source dynamics which

165 can be obtained from measurements with a micrometeorological method. It can alternatively be approached by

166 modelling using the state of the art of ammonia exchange processes as we do here.

167 Additionally to the bias, which is term (II) in **Eq. (4)**, evaluating term (I) is encompassed with errors related to

168 the uncertainties in $\overline{C_{bgd}}^{\tau}$, $\overline{C(x)}^{\tau}$ and $\overline{D(x)}^{\tau}$. In particular, cases when $\overline{D(x)}^{\tau}$ is small may lead to large errors in

169 inferring the source term $S$. This is linked to the conditioning of the inverse problem and is discussed in

170 supplementary material S2.

171 **2.1.3 Case of multiple sources and multiple concentration samplers with time averaging**

172 If we generalise the approach to multiple sources and multiple receptors, then the transfer function becomes a

173 matrix $D(x_i, S_j, t)$, which is the contribution of source $S_j$ to concentration at target located at $x_i$. For reading

174 purposes we simplify the matrix notation to $D_{i,j}$. **Eq (3)** then becomes:


176 $$\overline{\begin{bmatrix} C_1 \\ \vdots \\ C_M \end{bmatrix}}^{\tau} = \overline{\begin{bmatrix} D_{1,1} & \cdots & D_{1,M} \\ \vdots & \ddots & \vdots \\ D_{N,1} & \cdots & D_{N,M} \end{bmatrix}}^{\tau} \times \overline{\begin{bmatrix} S_1 \\ \vdots \\ S_M \end{bmatrix}}^{\tau} + \overline{C_{bgd}}^{\tau} + \overline{\begin{bmatrix} D'_{1,1} & \cdots & D'_{1,M} \\ \vdots & \ddots & \vdots \\ D'_{N,1} & \cdots & D'_{N,M} \end{bmatrix} \times \begin{bmatrix} S'_1 \\ \vdots \\ S'_M \end{bmatrix}}^{\tau} \qquad (6a)$$


178 Which in condensed notation gives:


180 $$\overline{C(x_i)}^{\tau} = \overline{D_{i,j}}^{\tau} \times \overline{S_j}^{\tau} + \overline{C_{bgd}}^{\tau} + \overline{D'_{i,j} \times S'_j}^{\tau} \qquad (6b)$$





If the number of targets is equal to the number of sources, the problem can be solved by inversion of a linear
system. If the number of targets is larger than the number of sources, the problem is a multiple linear regression
type with unknowns $\overline{S_j}^\tau$ and $\overline{C_{bgd}}^\tau$. The third term on the right hand side of the **Eq. (6b)** is a bias which is *a*
*priori* unknown and which we will evaluate in this study.

### 2.1.4 Source inference methods

The inferred sources, $\overline{S_i^{inferred}}^\tau$, were derived from **Eqns. (3)** or **(6)** assuming the covariance term (last term on
right hand side) was null. The method used to infer the source was either a simple division (**Eq. (3)**) or an
optimisation of the linear system using the linear model function *lm* in R (package stats, R version 3.2.3), with
either $M = 1$ (single source) or $M = 9$ (multiple sources):
$$\overline{\begin{bmatrix} D_{1,1} & \cdots & D_{1,M} \\ \vdots & \ddots & \vdots \\ D_{N,1} & \cdots & D_{N,M} \end{bmatrix}}^\tau \times \overline{\begin{bmatrix} S_1^{inferred} \\ \vdots \\ S_M^{inferred} \end{bmatrix}}^\tau = \overline{\begin{bmatrix} C_1 \\ \vdots \\ C_N \end{bmatrix}}^\tau - \overline{C_{bgd}}^\tau \qquad (7)$$

The bias $\delta S_i^\tau$ was then evaluated as the difference between the inferred sources $\overline{S_i^{inferred}}^\tau$ and the modelled
sources $\overline{S_i^{obs}}^\tau$ averaged over each period:
$$\delta S_i^\tau = \overline{S_i^{inferred}}^\tau - \overline{S_i^{obs}}^\tau \qquad (8)$$

As shown in **Eqns. (3)** and **(6)** the overall mean bias $\delta S_i^\tau$ contains (i) a bias term due to the inference method
which is dependent mainly on the conditioning of the matrix $D_{ij}$ (see supplementary material S2) and (ii) a bias
term which is intrinsically linked to the covariance between $D_{ij}$ and $S_j$ (**Eqns. 3** et **6**). Thus, with **Eq. (8)** we
evaluate the sum of the two biases without distinction. In order to infer the sources, the elements of the
dispersion matrix $D_{ij}$ need to be determined. The next part details how these were estimated with a dispersion
model.

### 2.2 The dispersion model used for determining the transfer matrix $D_{ij}$

The elements of the transfer matrix $D_{i,j} = D(x_i, S_j, t)$, were calculated using a dispersion model. Indeed, by
definition, $D(x_i, S_j, t)$ is the concentration at location $x_i$ and time $t$ generated by a source $S_j$ of strength $S_j(t) = 1$.
The FIDES-3D model ("FIDES",(Loubet et al., 2010), based on the analytical solution of the advection-diffusion
equation of Philip (1959) was used for that purpose. This model was first compared and tuned with a backward
Lagrangian Stochastic dispersion model (the "WindTrax" software, Thunder Beach Scientific, Nanaimo,
Canada, (Flesch et al., 1995). The two models and how the FIDES model was tuned are briefly described
hereafter and detailed in the supplementary material sections S3 and S4.
The FIDES model is based on the Philip (1959) solution of the advection-diffusion equation, which assumes
power law profiles for the wind speed $U(z)$ and the vertical diffusivity $K_z(z)$. This approach also assumes no
chemical reactions in the atmosphere and spatial horizontal homogeneity of roughness length ($z_0$), wind speed
($U$), vertical ($K_z$) and lateral ($K_y$) diffusivity. The dispersion model is detailed in Huang (1979), and Loubet



(2010). The details of the model and the way the transfer function $D(x_i, S_j, t)$ was estimated is detailed in the
supplementary material S2.
The Schmidt number which is the ratio of momentum to scalar vertical diffusivity $Sc = Km_z / K_z$ is key in
dispersion modelling, as it determines the vertical diffusion rate of scalars. Wilson (2015) demonstrated that bLS
and dispersion models like FIDES give different values of $Sc$ by constitution. In order to assure consistency of
the Phillip (1959) approach with bLS models, considered as references in dispersion modelling, we chose to tune
the Philip (1959) model to get the same $Sc$ number as in WindTrax as described by Flesch et al. (1995). The
details are given in supplementary material S4. The comparison showed that the tuned FIDES model gives very
similar concentrations to WindTrax at measurement heights lower than 2 m above the source, although slightly
overestimated under stable and neutral conditions and slightly underestimated under unstable conditions. The
correlation between the two models is however very high ($R^2 \geq \sim 0.96$) meaning that using the tuned FIDES
model to characterise source inference performance, will lead to results very similar to WindTrax. Moreover
since in this study the same model is used for predicting and for inferring the fluxes the results are self-
consistent.
**2.3 Ammonia sources from simple SVAT modelling and prescribed emission potentials**
In order to evaluate the bias introduced by time averaging the concentrations when inferring single or multiple
sources (third term in **Eqns. 3** and **6**), we generated $NH_3$ emission patterns mimicking the behaviour of real
sources as closely as possible. In that prospect, we used the SurfAtm-$NH_3$ model developed by Personne et al.
(2009), which was used for two purposes: (i) evaluating the turbulence parameters (the friction velocity $u_*$, and
the Monin Obukhov length $L$) from the meteorological datasets to parameterise the dispersion models, and (ii)
providing the surface temperature $T(z_0)$ and the surface resistances in order to calculate ammonia emission
patterns.
The SurfAtm-$NH_3$ model is a one-dimensional, bi-directional surface-vegetation-atmosphere-transfer (SVAT)
model, which simulates the latent ($LE$) and sensible ($H$) heat fluxes, as well as the $NH_3$ fluxes between the
biogenic surfaces and the atmosphere. It is a resistance analogue model separately treating the vegetation layer
and the soil layer, and coupling a slightly modified (Choudhury and Monteith, 1988) model of energy balance
and the two-layer bi-directional $NH_3$ exchange model of (Nemitz et al., 2000) with a water balance model.
Unless otherwise stated, the surface was considered a bare soil with $z_0 = 5$ mm, d = 0 m, and LAI = 0.
The ammonia emission patterns were modelled using the resistance approach and assuming atmospheric
concentration was zero, which is a reasonable assumption following nitrogen application and leads to patterns
mimicking reality, which is what we are seeking here:

$$F = \frac{C_{\text{pground}}}{R_a(z_{ref}) + R_b\{NH_3\}}$$
(9)


Where $R_a(z_{ref})$ is the aerodynamic resistance at the reference height $z_{ref} = 3.17$ m, and $R_b\{NH_3\}$ is the soil
boundary layer resistance for ammonia as described in Personne et al. (2009). The ground surface compensation
point concentration ($C_{\text{pground}}$) was expressed as a function of $\Gamma$, the ratio of $NH_4^+$ to $H^+$ concentration in the soil
water at the surface, as in Loubet et al. (2012):





$$C_{\mathrm{pground}} = K_h\{T(z_0)\} \times K_d\{T(z_0)\} \times \Gamma = \Gamma \times 10^{-3.4362+0.0508\,T(z_0)}$$ (10)


where $K_h$ and $K_d$ are the Henry and the dissociation constant for $NH_3$, and $T(z_0)$ is the soil surface temperature.
Since we wanted to evaluate the correlation between the transfer function $D_{ij}$ and the source strength $S_j$, which is
the bias in the inference problem (**Eq. 6**), the $NH_3$ volatilisation was modelled as to reproduce the variety of
existing kinetics of $NH_3$ emissions from fields. In that prospect, three $\Gamma$ patterns were simulated:

1. a constant $\Gamma = \Gamma_0$, which would mimic background $NH_3$ emissions from soils;

2. an exponentially decreasing $\Gamma = \Gamma_0 \exp(-4.6\,t/\tau_0)$, which best represents $NH_3$ emissions following slurry application;

3. a Gaussian $\Gamma = N(\Gamma_0,\sigma_\Gamma)$, which would represent the typical $NH_3$ emissions following urea application.

Here $\Gamma_0$ is the maximum $\Gamma$ during the period, $t$ is the time in days, $\tau_0$ is the duration of the emission in days. The
factor 4.6 was chosen so that when $t = \tau_0$, $\Gamma$ goes down to 1% of $\Gamma_0$. The duration of the emissions was chosen to
be four weeks, $\tau_0 = 28$ days. While these $\Gamma$ patterns gave the weekly trend of $NH_3$ emissions, the daily patterns
were produced by the thermodynamical and turbulence drivers of $NH_3$ emissions which were explicitly taken
into account through the compensation point (**Eq. 10**). To facilitate understanding, in most of the manuscript
only the constant $\Gamma$ was considered, and the effect of modifying the source strength was evaluated in a sensitivity
study.
**2.4 Spatial set up of the sources, concentration sensors**
The sources (plots) were considered as squares with width $x_{\mathrm{plot}}$ and aligned south-north. Two configurations were
considered: (1) a single source configuration and (2) a multiple-sources configuration which mimics typical
agronomic trials with 9 sources (plots) placed next to each other, with three treatments times three repetitions.
Each treatment was assigned a value of $\Gamma_0$ different from the others, while the three repetitions of the same
treatment were assigned the same value of $\Gamma$. The concentration sensors (receptors) locations, $x_i$, were set in the
middle of each plot, at several heights $z_i$. (**Figure 1**).

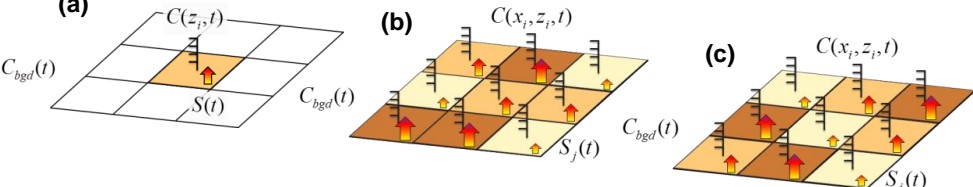


**Figure 1. General scheme of the source receptor locations for (a) a single source, and (b) multiple-sources. (c)**
**"optimum" plot layout used for the multiple-source configuration.**
A number of plot sizes ($x_{\mathrm{plot}} = 25, 50, 100$ and $200$ m on the side), and receptor heights ($z_i = 0.25, 0.5, 1$ and $2$
m), were tested successively. Several source strengths and dynamics were also tested: $\Gamma$ was first considered
constant with time (pattern 1) in all the plots , and the $\Gamma_0$ of each of the three treatments were either chosen to be
significantly different in strength ($10^4, 10^5, 10^6$), or of the same order of magnitude (1000, 2000, 4000). Then the
three $\Gamma$ patterns ("constant", "exponential" and "Gaussian") were randomly assigned to the treatments for each



simulation period. The ammonia background concentration, $C_{bgd}$, was considered constant and equal to 1 ppb
except when studying the sensitivity of the inference method to the background concentration, where it was set
as unknown. Throughout this study, an "optimum" block configuration was considered (shown in **Figure. 1c**),
which avoided trivial configurations like aligned blocks and maximised the mean distance between blocks.
**2.5 Simulation details**
**2.5.1 Meteorological data and fertiliser application periods**
A range of meteorological conditions were simulated based on the half-hourly meteorological data of the FR-Gri
ICOS site in 2008. In total 13 periods of 28 days were considered which spanned the whole year except the last
two days of the year. Each period consisted of 1344 half-hourly data.
**2.5.2 Concentration sensor integration periods**
In order to evaluate the influence of the concentration averaging period on the source inference, several
integration periods $\tau$ were tested: 0.5h (no integration), 3h, 6h, 12h, 24h, 48h, 168h (7 days). In practice the
concentrations were computed at each sensor location using **Eq. (6)** over 0.5h: at that frequency, the covariance
term is assumed to be negligible. Then the averaged concentrations were computed for all integration periods.
**2.5.3 Sensitivity to inferential methods hypotheses**
Several hypotheses were considered and summarized in Table 1:
1)   the background concentration $\overline{C_{bgd}}^{\tau}$ was either supposed known and fixed to the prescribed values (**C1-**

306        **C4**) or was inferred (**C5-C7**);

2)   the three repetitions of each treatment were either supposed to have the same source strength (**C2**, **C4**,

**C5**, **C6**) or they were inferred independently (**C1**, **C3**, **C7**). In C2, C4, C5 and C6, $S_i = S_m$ for all $i$ and

$m$ belonging to the same treatment. In practice a new dispersion matrix was calculated by averaging

together all columns belonging to the same treatment (matrix dimension $N \times 3$). Three strength values

of $S$ were inferred to be tested;

3)   either one concentration sensor at each source location ($z_i$) was considered (**C1**, **C2**, **C5**) or two sensors

positioned at two heights were considered (**C3**, **C4**, **C6**, **C7**). All the measurement heights and their

combinations were considered.


**Table 1. Hypotheses tested for inferring the sources and background concentration.**

| Strategy | Number of sensors | Plots[#] have same emissions | Background concentration | Note |
|---|---|---|---|---|
| C1 | 1 | No | known | Each block is considered independently |
| C2 | 1 | Yes | known | Each block is considered equal |
| C3 | 2 | No | known | Identical to C1 except for the number of sensors |
| C4 | 2 | Yes | known | Identical to C2 except for the number of sensors |
| C5 | 1 | Yes | unknown | Identical to C2 except for the background concentration estimation |
| C6 | 2 | Yes | unknown | Identical to C4 except for the background concentration estimation |
| C7 | 2 | No | unknown | Identical to C3 except for the background concentration estimation |

# plots are plots having the same treatment (repetitions).



**2.6 Statistical indicators**
For each run the mean bias (BIAS), normalised mean bias (NBIAS), were calculated as: $BIAS_i = \frac{1}{N_\tau}\sum_\tau \delta cumS_i^\tau$,
$NBIAS_i = BIAS_i / \left(\frac{1}{N_\tau}\sum_\tau cumS_i^{obs}\right)$, where $N_\tau$ is the number of the time averaged samples over each 28-day
period and $cumS_i$ and $cumS_i^{obs}$ are the cumulated fluxes over the same period. The medians and interquartile of
these statistical indicators were then calculated over the 13 periods of 28-days for 2008.
**2.7 Real experimental test case**
In order to evaluate the feasibility of the method, we applied it to a real test case (**Figure 2**). The trial was
located at La Chapelle Saint-Sauveur in France (47°26'44.1"N, 0°58'50.7'W',) and performed from 5[th] April to
26[th] April 2011 on a bare soil with loamy soil texture. Soil pH in water was 6.2 and the bulk density in the first
15 cm was 1.4 t m[-3].. The experimental unit consisted of 6 squared sub-plots of 20 m on each side with 2
repetitions of 3 treatments: (1) surface application of cattle slurry, (2) surface application and incorporation of
the same slurry and (3) no application. The slurry had a pH 7.5, a dry matter (DM) of 6.05%, C:N ratio of 10.4
and contained 38.45 g N kg[-1] (DM) as total nitrogen and 13.25 g N-NH$_4$ kg[-1] (DM) as ammoniacal nitrogen.
Slurry was applied on 5[th] April 2011 at a rate of 49 m[3] ha[-1] which led to 118.7 kg N ha[-1] and 40.9 kg N-NH$_4$ ha[-1].
The application was identical between the repetitions with a small standard deviation (< 0.2 kg N ha[-1]). The
incorporation was performed in two sub-plots one hour after the end of the slurry spreading with a disc harrower
at a depth of 0.10 m. The soil humidity between 0 and 5 cm depth was homogeneous over the blocks and
decreased from 20±1% to 17±1% w/w between the start and the end of the experiment. The meteorological data
were measured nearby (**Figure 2**). Air temperature, relative humidity, global solar radiation, wind velocity and
direction were recorded every 30 minutes at 2 m height. The dispersion model input parameters ($u_*$ and $L$) were
evaluated with a simple energy balance model of Holtslag and Van Ulden (1983) assuming a Bowen ratio of 0.5
and a deep soil temperature equal the averaged ambient temperature. Ammonia concentration was measured with
diffusive samplers (ALPHA samplers), (Sutton et al., 2001; Tang et al., 2001; Tang et al., 2009), which were
placed at the centre of each sub-plot at two heights (0.32 and 0.87 m from the ground) as well as next to the
assay at three location (5 m away from the plots) at 3 m height. The ALPHA samplers were set in place just after
slurry application and incorporation (between 14:20 and 14:50) and left exposed subsequently for 3 h, 22 h, 23 h,
23 h, 71 h (3 days) and 359 h (15 days) hence spanning 21 days. The diffusive samplers were prepared prior to
the experiment, stored at 4°C in a refrigerator and analysed by colorimetry. Since no background concentrations
were measured at a reasonable distance from the field, the background concentration was assumed as the
minimum over the whole period of the concentrations measured on the 3 m height masts.





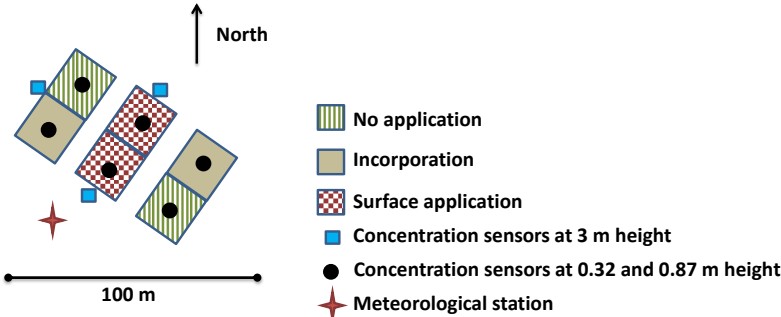


**Figure 2. Scheme of the real experimental test case performed on 6 sub-plots with three treatments and two repetitions. Cattle slurry was either applied on the surface or incorporated. The concentration sensor and meteorological station locations are shown on the scheme.**


**3  Results and discussion**
**3.1 Meteorological data range and simulated ammonia sources**
The meteorological conditions over the 13 periods represented a good sample of temperate climate conditions. In
particular $u_*$ and the stability parameter $z/L$ vary over each period and between periods from 0.024 to 1.181 m s$^{-1}$
for $u_*$ and from -49 to 21 m$^{-1}$ for $z/L$, respectively (**Figure 3**). It is noticeable that $u_*$ showed greater variability
during the winter than during the summer, while it was the opposite for z/L. The surface temperature also
showed a structure varying between periods, with a larger temperature range during the summer (from 5.7 to
50.4°C) than during the winter (from -5.2 to 22.9°C). This surface temperature variability is an essential feature
to representing real case ammonia sources (Sutton et al., 2009), which shows a variability reflecting both the
surface temperature and the resistances variations (**Eqns. 9 and 10**).





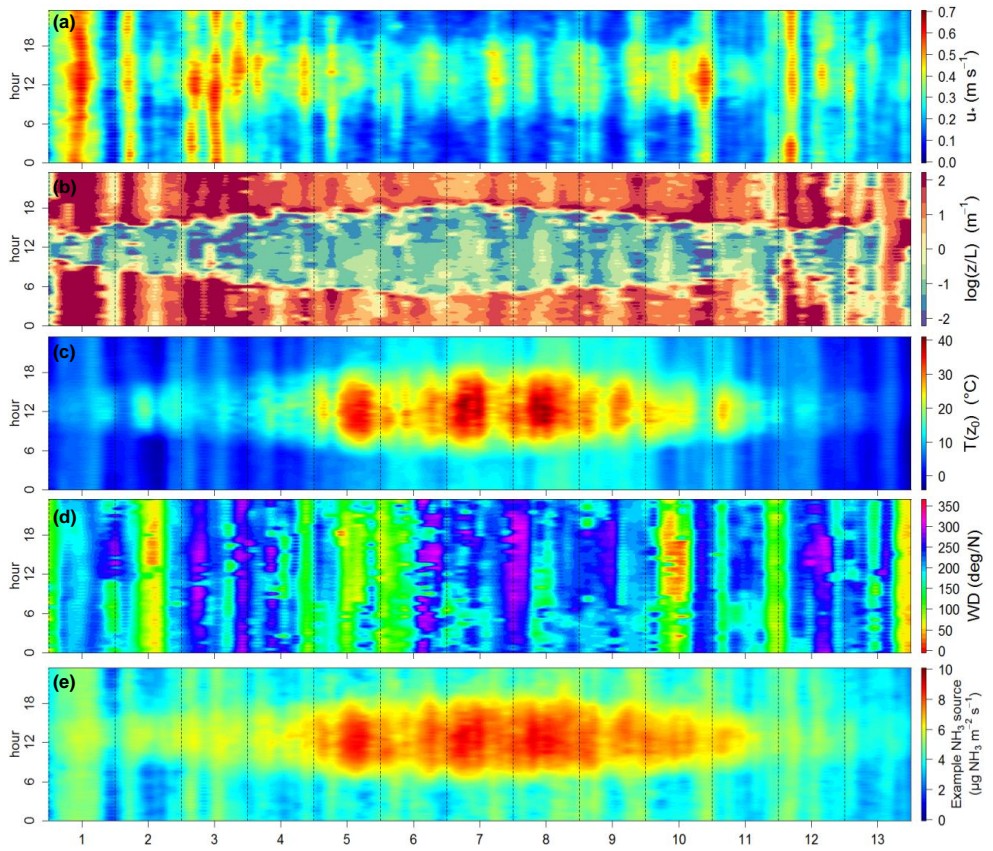


**Figure 3. Footprints of measured $u_*$ (a), $z/L$ at 1 m height (b), $T(z_0)$ (c), and wind direction (d) for the hour of the day**
**and the 13 considered periods over year 2008 in the FR-GRI ICOS site. The modelled ammonia source is also**
**reported (e) according to Eqns. (9) and (10) over the same period with a $\Gamma = 10000$.**


**3.2 Example ammonia concentration dynamics modelled with the tuned FIDES model**

The modelled ammonia concentrations reproduced typical patterns measured above field following nitrogen

application well, with maximum concentrations during the day and minimum concentrations at night (**Figure 4**).

These patterns are a consequence of daily variations of the sources driven by surface temperature combined with

variations in the aerodynamic transfer function $D_{ij}$, which behaves similarly as a transfer resistance (see

supplementary material S1). The integration periods are also shown in **Figure 4,** which illustrates the progressive

loss of information of the pattern structure with integration periods. Particularly, it can be seen that the day-to-

night variation is captured up to an integration period of 6h. Moreover, it should be noted that averaging also

means overestimating lower concentrations and underestimating higher concentrations.



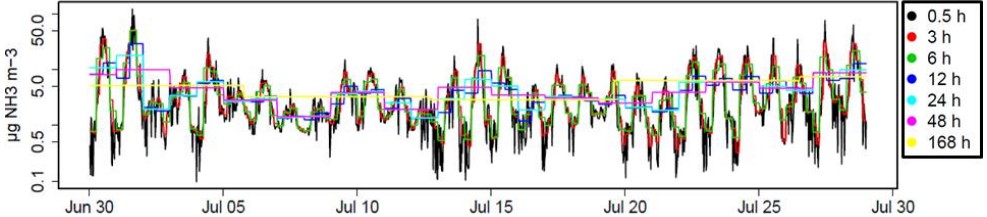

**Figure 4. Example modelled concentration pattern at 1 m above a single 50 m width source for several averaging**
**periods (0.5h – 168h) for the month of July 2008. The source $\Gamma$ was set to $10^5$. The y-axis is log scaled.**

### 3.3 Evaluation of the inference method for a single source and a single sensor

At first we evaluate the bias of the inference method for the simpler case of a single source and a single sensor
placed in the centre of the source field at several heights, assuming we know the background concentration
(strategy **C1**; **Figure 1a.**). This case has the advantage of having a condition number equal to 1 (**Eq. (S1)**) and a
bias $\delta S^\tau$ which is well defined and equal to $-[\overline{D^\tau}]^{-1} \times \overline{[D'S'^\tau]}$ (**Eq. (8)**). This section hence focusses on
evaluating the influence of sensor height, time integration, and source dimension on the bias without dealing
with the complexity of the interactions between multiple fields.

### 3.3.1 Example inferred source dynamics

**Figure 5** reports an example source inference, which shows the progressive smoothing of the source with
integration period. We first see that the source strength corresponding to $\Gamma = 10000$ leads to ammonia emissions
ranging from 0 to ~1 µg NH$_3$ m$^{-2}$ s$^{-1}$ in the winter, which corresponds to 0.71 kg N ha$^{-1}$ day$^{-1}$. Over the entire
year, the maximum emission occurs during the hottest days and reaches up to 7.1 kg N ha$^{-1}$ day$^{-1}$. Regarding the
inference method, it can be seen in that example that up to 24 hours the variability in emissions over the period is
captured quite well.





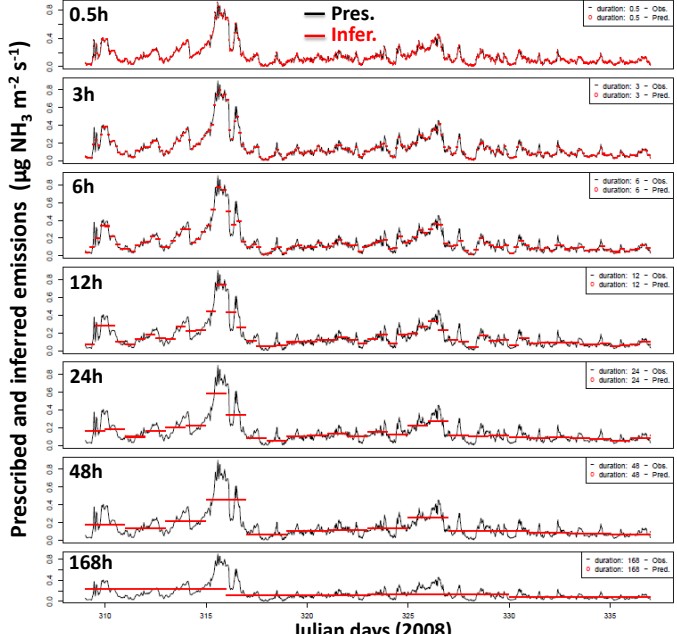

**Figure 5. Example source inference for a 25 m width square field and a concentration sensor placed at 0.5 m above ground. Here Γ = 10000 and is set to constant (pattern 1). The 7 integration periods are shown: 0.5h to 168h. The x-axis shows the day of year and corresponds to a span over November. The prescribed source is in black (Obs.) and the inferred one in red (Pred.)**

**3.3.2 Effect of target height, source dimension and integration period on the bias $\delta S^\tau$ for a single source**

In this simpler case shown in **Figure 6**, the fractional bias of the inferred emission is mostly negative for the combination where the ratio sensor height / plot dimension is small and integration times are larger than 6h. According to **Eq. (5)**, this means that the covariance term $\overline{D'S'^\tau}$ is negative for these conditions, meaning that any increase in source strength $S$ at a time $t$ is correlated with a decrease of the transfer function $D(t)$ and vice versa. This is expected as $S(t)$ increases with the surface temperature (**Eq. (10)**) and is proportional to $[R_a(z_{ref}) + R_b\{NH_3\}]^{-1}$ (**Eq. (9)**), while $D(t)$ is proportional to the aerodynamic resistance $R_a(z_{\text{ref}})$, as shown in supplementary material S1. Hence, over daily periods, $S$ and $D$ are negatively correlated: $S$ increases during the day and decreases at night (due to temperature and wind speed daily patterns), while $D$ decreases during the day and increases at night (mainly due to wind speed patterns). This is expected to be a general feature for NH$_3$ surface fluxes as the daily variability reproduced by the model used in this study is representative of most situations from mineral and organic fertilisation, to urine patches or seabird colonies (Ferrara et al., 2014; Flechard et al., 2013; Milford et al., 2001; Moring et al., 2016; Personne et al., 2015; Riddick et al., 2014; Sutton et al., 2013).

The median bias $\delta S_i^\tau$ tends to increase in magnitude with the sensor height for large fields (100 and 200 m on side) whilst decreases for smaller fields (25 and 50 m on side) when sensor height gets close to the field boundary layer height. Furthermore, $\delta S_i^\tau$ becomes positive and very large when sensors get above the field





boundary layer height (**Figure 6**). For large fields, the increase of the magnitude of the bias with lower sensor
height is expected as $D$ decreases with height in absolute value. For small fields, the decrease of the bias
corresponds to a loss of information as $D$ gets close to zero when the sensor gets closer to the field boundary
layer height. For heights above this limit, we observe a change in sign of the bias which can be explained by the
fact that the sensor concentration footprint is not in the source during stable conditions (at night) while it is in the
source under unstable conditions during the day. The inference method will hence not work if at least one sensor
is not below the plot boundary layer height.

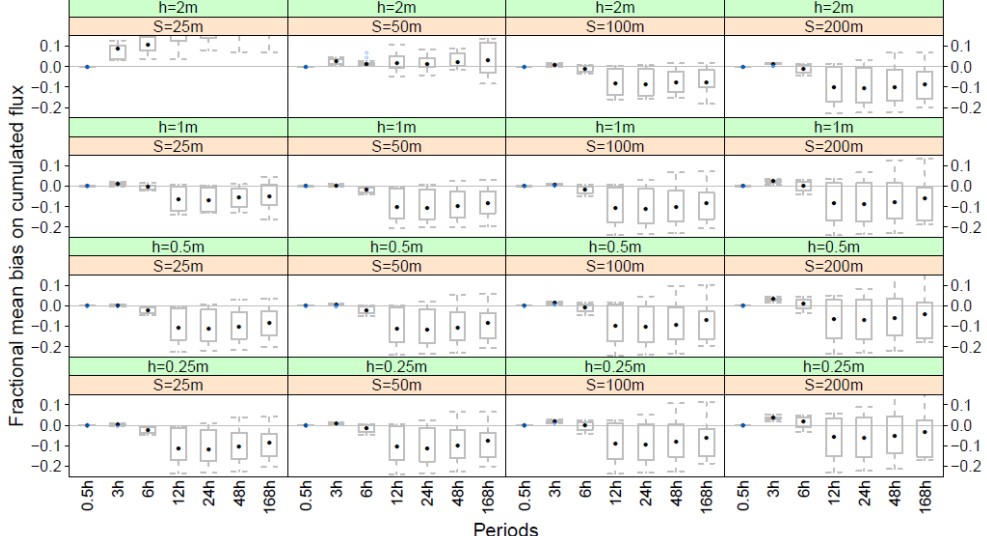

**Figure 6. Fractional bias of inferred cumulated ammonia emission for a single squared field of side (S) 25, 50, 100 and**
**200 m and sensors heights (h) 0.25, 0.5, 1 and 2 m, as a function of the sensors integrating periods from 3 hours to 1**
**week (168h). The points show the median, the boxes the interquartile and the whiskers the maximum and minimum**
**over the 13 application periods.**

We also notice that for integration periods below 3h, the fractional bias is slightly positive, which can be
explained by the positive correlation between $S$ and $D$ at small time scales. This is because of the influence of $u_*$
on $T(z_0)$: for a given solar radiation and air temperature over small time scales (< 3h), an increase in $u_*$ leads to a
decrease in $T(z_0)$, which leads to an exponential increase of the surface compensation point according to **Eq.**
**(10)**. However, at the same time, $R_a(z)^{-1}$ decreases, but linearly with $u_*$. The resulting ammonia emission
calculated with **Eq. (9)** nevertheless increases because the exponential effect of temperature overcomes the linear
effect of the exchange velocity (data not shown). This effect is more visible for large fields than small fields
because over small fields an additional effect is that when $u_*$ decreases, the footprint increases and the source
"seen" by the targets hence decreases because it incorporates a fraction of zero emission sources.
Overall, the median fractional bias for weekly integrated emissions over a 25 m field and sensor heights below
0.5 m was overall -8% with an interquartile (-14% to -2%). We can conclude that the bias of the $NH_3$ emissions
is reproducible within ± 6%. We can also conclude that it would be better to place the concentration sensor at a
low height to minimise the bias of the method.




### 3.3.3 Effect of surface boundary layer turbulence on the inference method for a single source

The inference method depends on the turbulence at the site and especially on the main drivers of the dispersion which are the friction velocity and the stability regime. Indeed **Figure 7** shows that the relative root mean square residual of the inferred source (RRMSR) decreases with increasing $u_*$ at long integration periods and is larger in slightly stable than near-neutral or slightly unstable conditions. **Figure 7** also shows that the under stable conditions or low $u_*$ the RRMSR increases by more than an order of magnitude (up to 50%) when integration periods increase from 6 to 12 hours, which catches most of the source variance. We also see that under near-neutral or high $u_*$ conditions, the 3$^{rd}$ quartile of the RRMSR remains below 10% for all integration periods. Finally, we also see that the larger 3$^{rd}$ quartiles at short integration periods are obtained with intermediate $u_*$ values or slightly unstable conditions. A similar response of the bias to $u_*$ and $1/L$ was reported by Figure 6 in (Flesch et al., 2004) and Figure 3 in Gao et al. (2009) in controlled source experiments. While Gao et al. (2009) attributed the bias of the inference method to parameterisation of the stability dependence of the turbulent parameters ($z/L$), in this study this cannot happen since we use the same parameterisation for prescribing the concentration and inferring it. In our case, the interpretation is to be linked with **Eq. (5)**: the smaller $u_*$ or the most stable conditions also correspond to the larger time-derivatives of source strength (driven by surface temperature and surface exchange resistances) as well as the larger time-derivatives of transfer function $D$. We hence expect that under such conditions, the covariance between the transfer function and the source strength will be larger than under near-neutral conditions. In a more heuristic view, under low turbulence, large time-derivatives of concentrations are expected above a source due to low mixing (small changes in mixing lead to large variations in concentrations).

We conclude that the inference method with a long integration period will lead to very moderate biases for locations with near-neutral conditions and high wind speed, typical of oceanic climates, but may lead to much larger bias under stable conditions and low wind speed typical of continental climates, as soon as the integration period gets up to 12 hours.



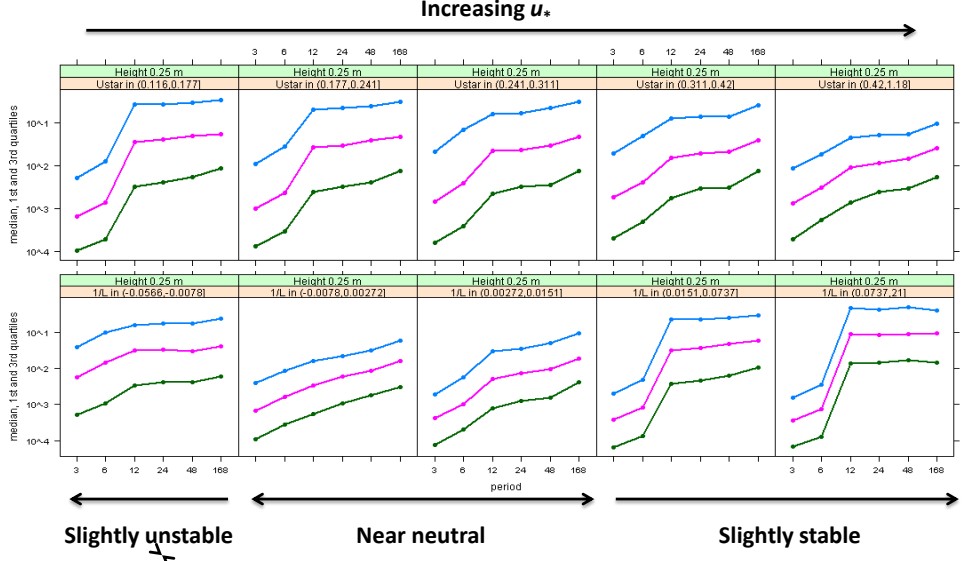

**Figure 7. Source relative root mean squared error as a function of integration period for stability factor and friction velocity classes for a single 25 m side field. Medians and quartiles are given for equally sized bins of $u_*$ and $1/L$ and for the lowest sensor height (0.25 m). The blue, pink and green curves are the 3rd, 2nd and 1st quartiles, respectively.**

### 3.4 Multiple source case

In contrast to the single source case, with multiple sources (see **Figure 1b**) the inference method leads to biases at small integration times as can be seen in the example reported in **Figure 8**. In that specific case, the emissions of treatments 2 and 3 are 10 times and 100 times larger than that of treatment 1, respectively. This leads to concentrations over plots of treatment 1 (and to a lesser extent over those of treatment 2) being highly correlated to emissions from plots of treatment 3 (and hence less with sub-plots of treatment 1). As a result, inferring emissions of plots of treatment 1 becomes harder as soon as averaging periods become larger or equal to 3h. This can be viewed as a progressive loss of information of the treatment 1 contribution to concentrations due to the overweighing contribution of treatment 3 plots. However, we also see that treatments 2 and 3 seem quite correctly inferred for integration times smaller than 48h.





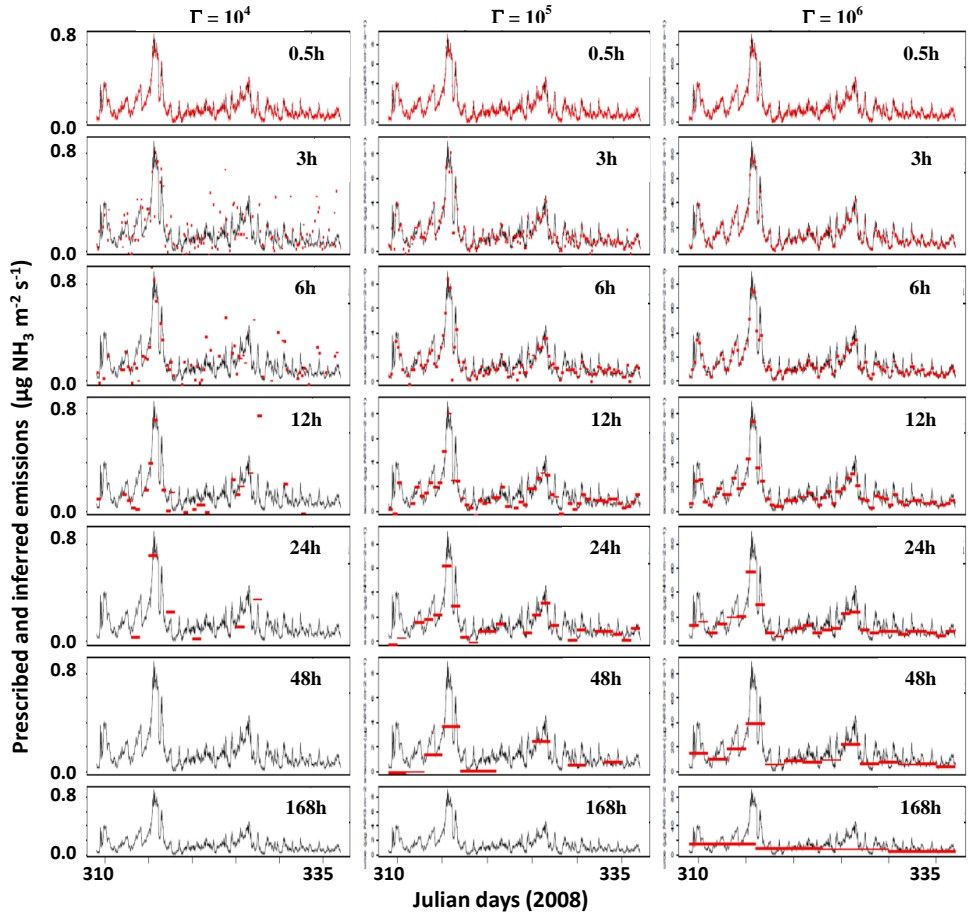

**Figure 8. Example result of multiple plot case inference. Black curves: observations; red dots: inferred sources. Left: $\Gamma = 10^4$. Middle: $\Gamma = 10^5$. Right: $\Gamma = 10^6$. Missing red dots are out of the y-scale boundaries. Example plots from treatments 1, 2 and 3 are shown from left to right. The period is the same as in Figure 7 (November 2008 for the FR-Gri ICOS site), and emissions are up to 1, 10 and 100 μg NH$_3$ m$^{-2}$ s$^{-1}$, for the three emission potentials. Strategy C7 with target heights 0.25 and 2 m, and source width 25 m on a side.**

In the following we will first evaluate the influence of the length of integration periods, sensor heights and plots dimensions on the fractional biases made when inferring the source. Each factor will be evaluated independently of the others in order to understand the processes behind it. For these evaluations background concentration was kept constant at 1 μg NH$_3$ m$^{-3}$. Strategy C1 was used except when testing sensor heights for which strategy C3, which uses two targets, was also used. These two strategies assume that the background concentration is known which avoids any compensating effects between source and background concentration inferences. Then the sensitivity of the methodology to the (i) emission ratios between two of the three treatments and (ii) the variability in the background concentration were evaluated. Finally, seven inversion strategies were compared to determine which was the most robust (**Table 1**).





### 3.4.1 Effect of integration periods on the bias


We first consider strategy C1, which is the simplest configuration, in which plots are independent, background
concentration is known and one target is used above each plot. **Figure 9** shows that for the given treatment
range (~1-10-100 µg NH$_3$ m$^{-2}$ s$^{-1}$), the fractional mean bias is lower than 0.2 in magnitude for the treatment
emitting the most (treatment-3), lower than 0.4 for the intermediate treatment (treatment-2) and up to 8 for the
treatment emitting the least (treatment-1); here we considered the 0.25-0.75 quantiles. The bias of the highest
treatment (treatment -3) actually behaves similarly to a single source case (**Figure 6**), with a median bias around
10% for 48h integration periods. This is expected because treatment-1 and treatment-2 have much smaller
emission strength and hence little influence on the concentration above the treatment-3 plots, which therefore
behaves in a similar manner to a single source. As a consequence, this bias in treatment-3 is mainly due to the
anti-correlation between $D$ and $S$ which increases with integration periods. The fractional bias is very large for
treatment-1 even for small integration periods. The bias can either be positive or negative showing that this
method does not allow for a correct estimation of the smallest sources.

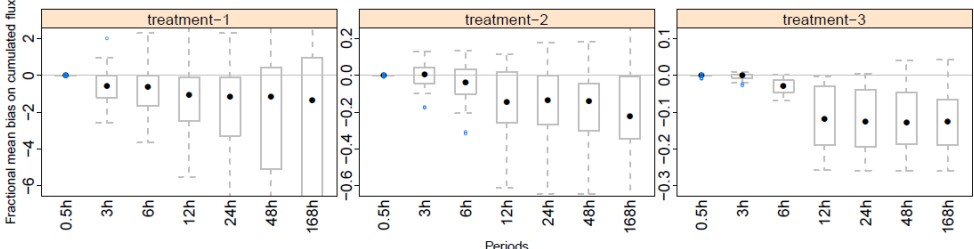


**Figure 9. Effect of integration period on source inference in a multiple-plot setup. The fractional mean bias of the**
**source is shown for each treatment (1 to3) corresponding to Γ = 10$^4$, 10$^5$, 10$^6$. Inference strategy C1 was used (single**
**sensor, independent blocks, background concentration known). Statistics for runs with target heights 0.25 and 0.5 m**
**and source side = 25 m are calculated. All application periods are considered. Filled points show medians, boxes show**
**interquartiles and bars show minimums and maximums. Outliers are points to 1.5 times away from boxes limits.**


### 3.4.2 Effect of target heights on the bias

**Figure 10** shows that the bias remains quite stable as long as sensor heights are low enough to catch a sufficient
part of the field footprint. When only a single height is used (strategy C1) this means that the sensor should be
placed at 0.5 m or below for the field size we have tested here (25 m), while for a pair of sensors (strategy C3)
the bias remains stable even for sensors places above 0.5 m.






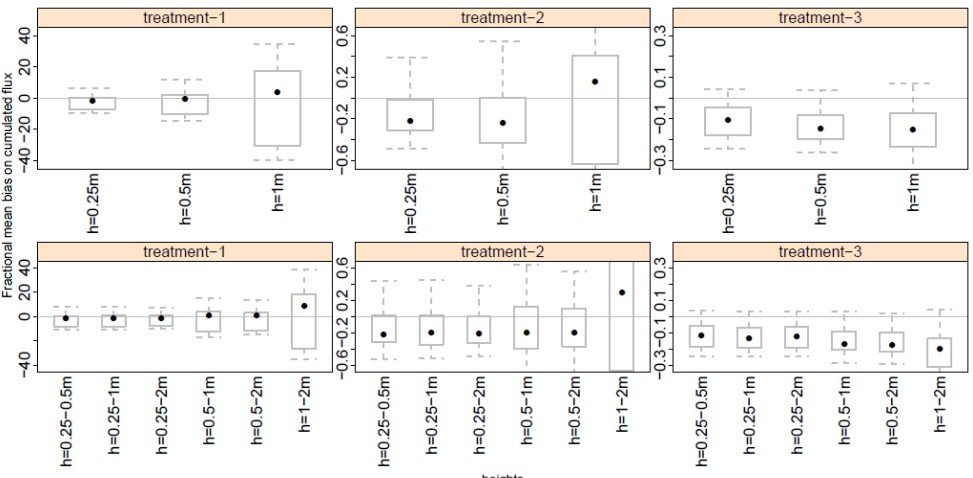


**Figure 10. Effect of target heights on source inference in a multiple-plot setup for integration periods of one week (168h). Same as the case reported for Figure 8 except that strategies C1 (with a single sensor) and C3 (with two heights) are compared here (the background is assumed known in both strategies).**


### 3.4.3. Effect of plot size on the bias

Increasing the plot size from 25 to 200 m width reduces the bias of the two largest source treatments for which the median bias reaches values around 10%, while the interquartiles remain stable (**Figure 11**). On the contrary, in treatment-1 (the lowest source), the bias increases. It is expected that the bias in a multiple-source configuration never becomes smaller than the bias in a single source problem which is a limit linked to the time-integration (covariance between the source and the concentration, see **Eqns. (3) and (6)**. It is also expected that the biases remain higher than the single source case until the source size increases sufficiently so that the concentration generated by a block on the neighbour fields become negligible compared to the concentration generated by the source below. This is what we observe in treatment-2 and treatment-3, withtreatment-2 showing a median bias of -13% (larger than in the single source case) for the 200 m large field, while the bias of the highest source tends to be -10% [-17%, -1%], which is the range observed for a single source.

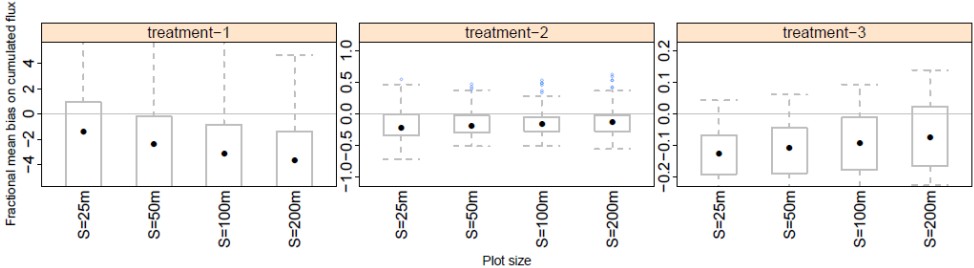


**Figure 11. Effect of plots size on source inference in a multiple-plot setup for integration periods of 168h and target heights 0.25 and 0.5 m. Same as in Figure 8.**






### 3.4.4 Sensitivity of the method to ratios of emission potentials between treatments

A central question is the capability of the inference method to resolve small or large differences in emissions
from the nearby blocks. Indeed, we can speculate that small differences will be hard to resolve while large
differences will lead to large bias. In order to determine the resolution power of the method, we compared the
performance of the inference method with a set of three treatments: the first treatment had $\Gamma = 0$ to mimic a
reference field receiving no nitrogen. The second treatment had a constant $\Gamma = 1000$ corresponding to a small
emission (0.7 kg N ha$^{-1}$ day$^{-1}$), while in the third treatment $\Gamma$ was successively set to increasing values from 1500
to $10^5$ (70 kg N ha$^{-1}$ day$^{-1}$). In this section we consider that the background is known (sensitivity to the
background concentration will be evaluated in the next section).
**Figure 12** shows the median and interquartile biases of the cumulated emissions for the longest integration
period 168h over the ratio of the high-to-low source treatments. The bias of the highest source always remained
around 14%, which is larger than the single source case. The bias of the lowest source increased with increasing
inter-treatments source ratio from 13% to 40%. In fact we find that the fractional bias increased approximately as
a power function of the ratio of the two predicted sources (dotted lines, $0.11\ x^{0.256}$).

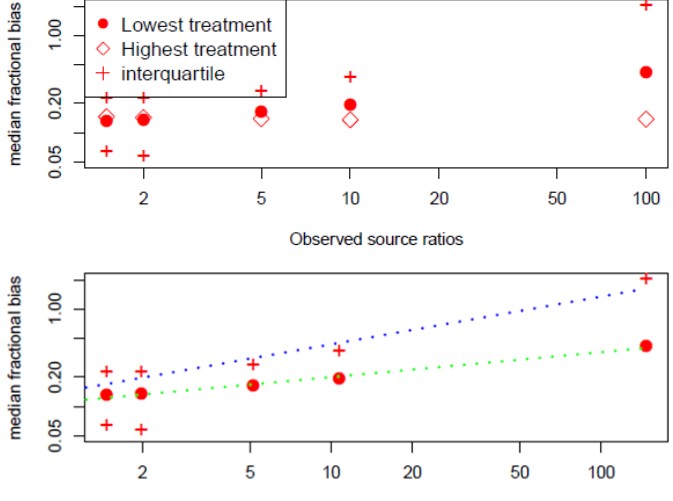

**Figure 12. Right: Median fractional bias of cumulated emissions as a function of the ratio of the high-to-low source**
**treatments for a 7 days integration period. Top: bias as a function of the theoretical source ratios. Bottom: bias as a**
**function of the predicted source ratios. Dotted lines show power functions regressions on medians (green) and**
**interquartile (blue). Strategies C1 and C3 are pooled together with all runs including sensor heights 0.25 and 0.5 m**

### 3.4.5 Quality of background concentration estimations

As pointed out by Flesch et al. (2004), the knowledge of the background concentration is essential in a source
inference problem. Retrieving the background necessitates having at least $N_{sources}$+1 sensors. Hence only
strategies with two heights per plot or which assume identical emissions in treatment repetitions can be evaluated



in their capacity of retrieving the background (strategy C2 to C7).  In order to evaluate the sensitivity of the
method when the background concentration varies with time, we set a realistic background concentration as a
linear combination of $u_*$ and air temperature ($T_a$) with a mean of 6 µg NH$_3$ m$^{-3}$, and a standard deviation of
0.1 µg NH$_3$ m$^{-3}$. This test was performed with a range of treatments in order to elucidate the correlations between
varying background and varying treatments. We see in **Figure 13** that the concentration, which follows a
realistic pattern, is well retrieved, even over the longest period. However, we see that for the treatments with the
largest source contrast ($\Gamma = 1000$ and $10^5$), the background concentration can be overestimated even for small
integration periods (6h). The median residual of the background concentration was smaller in magnitude than
0.05 µg m$^{-3}$, except for the case with very large differences between treatments (0, 1000, 10000), for which the
residual reached 0.1 and 0.5 for the 6h and 24h/168h integration periods. Furthermore, the background
concentrations were overestimated for the largest source ratios and underestimated for the lowest source ratios
and longer integration periods (24h and 168h).

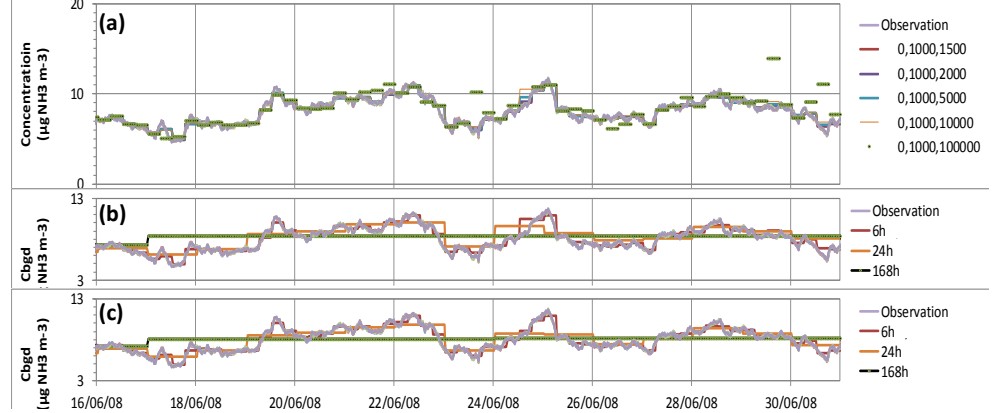

**Figure 13. Background concentrations prescribed (Observation) and inferred using strategy C7 and height**
**combination (0.25 m, 2 m): (a) effect of the treatment contrasts for a short integration period of 6h (treatments 1, 2**
**and 3 are given); (b) effect of integration period for contrasted treatments ($\Gamma = 0$, 1000, 10000); (c) effect of integration**
**period for similar treatments ($\Gamma = 0$, 1500, 2000).**

**3.4.6 Identifying the most robust strategy**
Finally to identify which strategy is the most suitable for retrieving the emissions, we compare all strategies on a
simulation with a variable background (set as in the previous section) and two sources ratios of 2 and 20 between
treatments 2 and 3 (treatment 1 being a zero source reference). We found, as expected, that strategies with
known backgrounds have low biases compared to strategies that calculate the background except for the strategy
C7 which provided biases similar to strategy C3 which is the strategy equivalent to C7 but with known
background (**Figure 14**). We also see that incorporating some knowledge of the sources by assuming plots from
the same treatment have the same emissions, gave slightly better estimates when the background is known
(strategies C2 and C4 compared to C3). This is however not true when the background is unknown, in which
case the magnitude of the bias increases up to a median of 0.7 (strategies C5 and C6 compared to C7). It is due to
compensation between background concentration and source strength as we have seen in **Figure 14,** that the



background concentration was overestimated in such cases. We also see, as expected, that the strategies with two
sensors placed at different heights above each plot lead to better evaluations of the emissions. Overall, the
strategy based on two sensors above each plot, which also assumes that sources are independent, seems to be the
most robust (strategy C7). This strategy does not assume the background is known, nor does it assume the plots
have similar emissions, which is more adapted to reality. Indeed, even though the same amount of nitrogen is
applied in each repetition plot, the emission may vary due to soil heterogeneity and advection. We finally get a
median bias for strategy C7 which is -16% with an interquartile [-8% -22%]. It is important to stress though that
the minimums and maximums are further away, which indicates that under some rarer circumstances, the method
may overestimate the sources by 12% or underestimate them by 40%. These cases correspond to integration
periods with very low wind speeds and stable conditions.

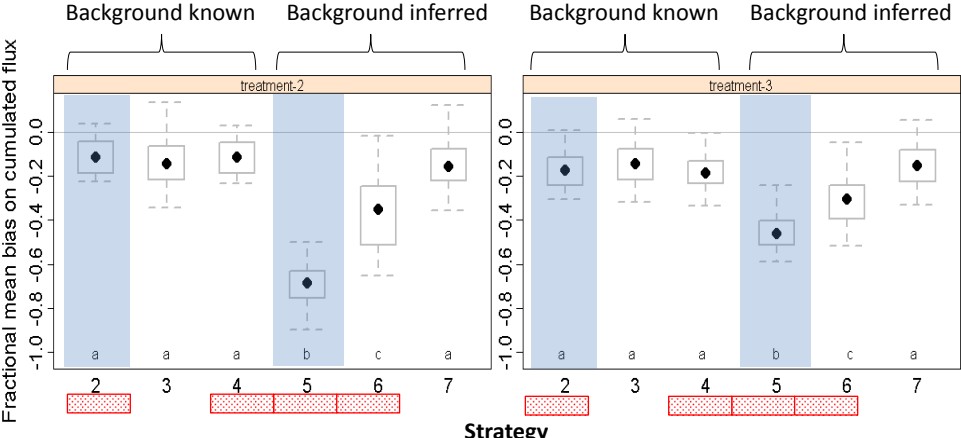


**Figure 14. Comparison of biases for all source inference strategies. In strategies C2, C3 andC4 we hypothesize that we**
**have perfect knowledge of the background concentrations, while in strategies C5, C6 and C7 background**
**concentrations are inferred together with the sources. In strategies C2, C4, C5 and C6 (red rectangles) we suppose**
**that plots from the same treatment have the same emissions, while in strategy C3 and C7 we infer each plot**
**separately. In strategy C2 and C5 we assume single sensors are placed above each plot (blue shades), while in**
**strategies C3, C4, C6, C7 we assume two sensors are placed above each plot.**

**3.5 Application of the methodology to a real test case with multiple treatments**
The evaluation of the methodology on a real test case is shown in **Figures 15-17**. The concentration measured
above the different treatments shows a much higher concentration above the surface applied slurry (up to
200 μg N-$NH_3$ m$^{-3}$) than above the two other treatments (below 50 μg N-$NH_3$ m$^{-3}$), (**Figure 15**).



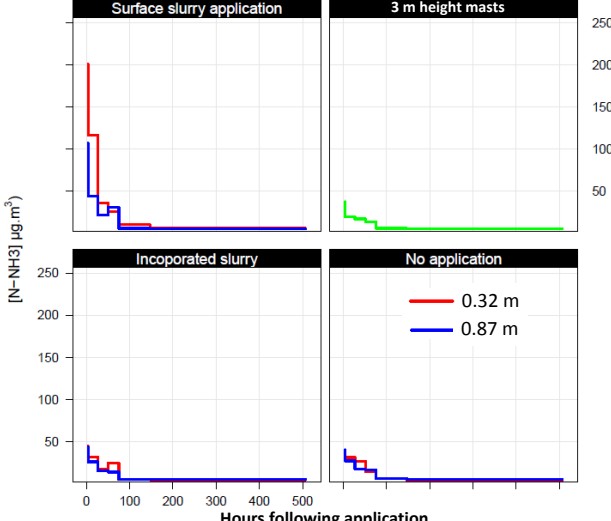


**Figure 15. Concentrations measured in a real test case with 6 blocks composed of three treatments and two repetitions. Here the mean concentration for the repetition and the three replicates ALPHA samplers are shown at two heights above ground. The concentration measured at 3 m height at 5 m away from the plots is also shown in green. The background concentration, evaluated as the minimum of the green curve was 5 µg N-NH$_3$ m$^{-3}$.**

The inference method gives very consistent results both in terms of comparison between repetitions of a given treatment and in terms of comparison between treatments (Strategy C7 shown in **Figure 16**). Emissions above 3 kg N ha$^{-1}$ (3.2 kg N ha$^{-1}$ on average) were found for the surface slurry application with a very good reproducibility between repetitions. This corresponds to an emission factor around 8.2% of the N-NH$_4$ applied and 2.8% of the total N applied, which is in-line with agronomic references (Sintermann et al., 2011a; Sommer et al., 2006). In contrast, the incorporated slurry showed much smaller and more variable fluxes between -0.2 and 0.25 kg N ha$^{-1}$. Furthermore, no significant differences were found between the no-application and the slurry incorporation treatments (Student t-Test with p-values larger than 0.03).


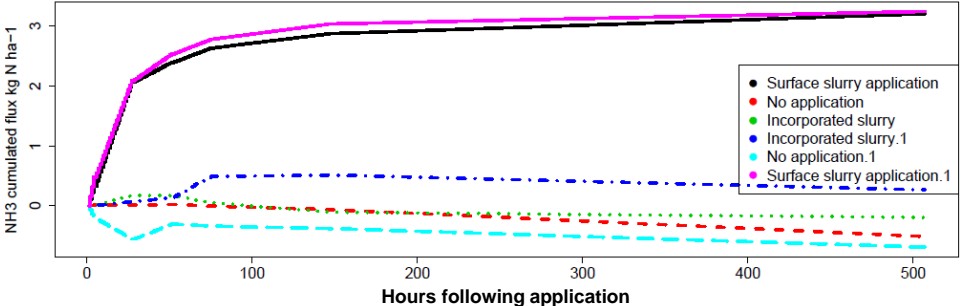


**Figure 16. Cumulated fluxes estimated with the inverse method on the real test case with strategy C7. Three treatments with two repetitions are compared.**






Comparing the inference strategies is instructive (**Figure 17**). We see that in methods which assume a known
background (strategies C3 and C4), the inferred emissions are higher than when background is assumed
unknown. We should remind that we set the background concentration to the minimum concentration measured
on the 3 m height masts because these were located too close to the plots to be considered as real background
masts. This explains why strategies C3 and C4 lead to higher estimates compared to strategies C6 and C7, as the
background may have been underestimated. We also find that all methods consistently infer a deposition flux to
the blocks with no application, which is consistent with our knowledge of ammonia exchange between the
atmosphere and the ground (Flechard et al., 2013). Indeed, the concentration in the atmosphere, which is
enriched by the nearby sources is expected to be higher than near the ground, due to a low soil pH (6.1), a low
nitrogen content in the soil surface (6-9.5 g N kg$^{-1}$ DM), and a 20% humid soil surface, hence leading to a flux
from the air to the ground.

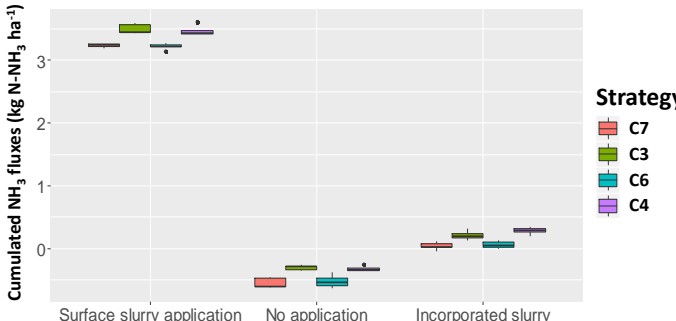


**Figure 17. Same as Figure 14 but grouped by treatments and with additional strategies C4 and C6 which consider that**
**replicates have the same surface flux. The variability in the boxplot aggregates the uncertainty on the inference**
**method (the standard deviation on the flux estimate in the least-square model, which accounts for the variability in**
**the replicated concentration measurements), and the variability between the repetitions in each treatment.**
From our theoretical study we know that strategy C7 should give a bias around -15 ± 8%. Therefore, we could
expect that the real flux is the one measured with C7 times (1.15 ±0.08), hence 3.7 ± 0.25 kg N ha$^{-1}$. This
corresponds to 10.1 ± 0.7 % of the N-NH$_4$ applied and 3.4 ± 0.2% of the total N applied. For the incorporated
slurry, the uncertainty would be much larger and is not evaluated here. We should bear in mind that the
theoretical study is based on the median of the simulations done with the 2008 dataset in Grignon which had
similar meteorological conditions to this trial. It would be much more relevant in future developments to
evaluate the bias based on the same method as developed here but based on the emissions and meteorological
conditions of the real case.
**3.6 Comparison with previous work**
Several studies have reported methodologies for evaluating multiple sources using dispersion models. These
were mostly based on Backward Lagrangian modelling (Crenna et al., 2008; Flesch et al., 2009; Gao et al.,
2008). There were several inference methods reported: the methods based on the inversion of the dispersion
matrix $D_{ij}$ or singular value decomposition of least-square optimisation (Flesch et al., 2009), which optimise the
conditioning of the dispersion matrix and one based on Bayesian inference (Yee and Flesch, 2010). Yee et al.
(2010) showed that the Bayesian approach would avoid unrealistic source estimates which could appear when




the matrix conditioning was poor. Unrealistic source estimates were for instance reported by Flesch et al. (2009),
with negative emission sources.
In Ro et al. (2011), they evaluated the bLS technique to infer two controlled methane surface sources with laser
measurements. They found 0.6 recovery ratios (ratio of inferred to known source) if the fields were not in the
footprint of the sensor but with adapted filters, they found a high degree of recovery with of $1.1 \pm 0.2$ and
$0.8 \pm 0.1$ for the two sources respectively. They found that in contradiction to Crenna et al. (2008) and Flesch et
al. (2009), even with large conditioning numbers they had high recovery rates.

### 3.6.1 Sensor positioning and conditioning number

Crenna et al. (2008) have clearly shown that the optimal sensor positioning should be so that each sensor sees
preferentially a single source, and reversely, each source should preferentially influence a single sensor. In this
study the sources-sensors geometry was especially designed in a way that minimises the condition number $CN$,
by placing the sensors in the middle of each plot. For the smallest source (25 m width), the conditioning number
ranged from 1.97 to 3.01 (median 2.42) for sensors located at 0.25 m, and increased  to 2.6-6.9 (median 3.2) for
sensors at 0.5 m, 4.7-150 (median 21) for sensors at 1.0 m, and 40-165000 (median 640) for sensors at 2 m. This
shows that including at least one sensor per block at heights lower than the field width divided by 20 would
ensure that the conditioning number remains lower than in most trials of Crenna et al. (2008).
By comparing different strategies we have found that the strategies using two sensors over each source
systematically led to improved performances (C3 versus C1 and C6 versus C5, **Figure 14**). This is also in line
with the results of Crenna et al. (2008), who showed that using more sensors separated spatially improves the
performance of the inference method. Hence we can conclude that the inference method we used is based on a
well-conditioned system which leads to robust results of the least-square optimisation. This is further illustrated
by the real case example (**Figures 15-17**) which shows a good reproducibility between block repetitions. Indeed,
good reproducibility between repetitions is a check for evaluating the quality of the inference method in real test
cases. The use of Bayesian inference method would however also be valuable in the setup we propose here.

### 3.6.2 Effect of time integrating sensors on the source inference quality

The use of time averaging sensors for estimating ammonia sources was already reported by Sanz et al. (2010),
Theobald et al. (2013), Carozzi et al. (2013a; 2013b), Ferrara et al. (2014) and Riddick et al. (2016a; 2014). All
these studies have shown the feasibility of these measurements, however only a few of them allow estimating the
impact of averaging: Riddick et al. (2014) measured emissions from a bird colony in the Ascension island with
WindTrax using both several alpha samplers in a transect across the colony and a continuous analyser for
ammonia (AiRRmonia, Mechatronics, NL) downwind. They also averaged the continuous sampler
concentrations to evaluate the effect of averaging on the emissions estimates. They found as we do here that
averaging over monthly periods would lead to systematic underestimations from -9% to -66%.  They also found
that estimations from badges would lead to average underestimations of -12%. This is very close to what we find
here for a single source over one week (**Figure 6**). In a similar comparison Riddick et al. (2016b) found that
time-integration led to slight overestimations with integration approach, which is within the range of statistics of
the bias we have found for the larger area sources (3rd Quartile in **Figure 6**).



### 3.6.3 Dependency to meteorological conditions


We should bear in mind that the use of time averaging sensors in the inference method is also highly dependent
on the surface layer turbulent structure as shown by **Figure 7**. We find, as expected, that stable conditions or low
wind speed conditions are those that lead to the highest potential bias (as shown by the 3[rd] quartile under stable
conditions in **Figure 7** bottom). This is a well-known limitation of inverse dispersion modelling which was
reported by Flesch et al. (2009; 2004) and which suggested that inverse dispersion would be inaccurate for
$u_*$ < 0.15 m s[-1] and $|z/L|$ < 1. However, both our study and the studies of Riddick et al. (2014; 2016b) show that
this is not as much of an issue for ammonia emissions. Indeed, this is due to the fact that ammonia emissions
follow a daily cycle with low emissions at night and high emissions during the day. This is firstly because (1) the
ground surface compensation point concentration ($C_{\mathrm{pground}}$) has an exponential dependency on surface
temperature as assumed in **Eq. (10)** based on known thermodynamical equilibrium constants (Flechard et al.,
2013). This is secondly due to the fact that ammonia emission is a diffusion-based process which is limited by
the surface resistances, as modelled in **Eq. (9)**, which leads to small fluxes when $R_a(z_{ref})$ and $R_b\{NH_3\}$ get
large, which happens during low wind speeds (they are both roughly inversely proportional to wind speed) and
stable conditions, which also happens at night (Flechard et al., 2013). In real situations, the combination of small
turbulence and high surface concentration leads to a further decrease of the flux which is dependent on the
difference between $C_{\mathrm{pground}}$ and the concentration in the atmosphere above (a feature which was not accounted
for in this study as this would imply a higher degree of complexity in the modelling approach). This means that
the results we found in this study would not apply for species having an emission pattern with a different
temporal dynamics (either constant or anti-correlated with surface temperature or wind speed).

### 4. Conclusions


In this study we have demonstrated that it is possible to infer with reasonable biases ammonia emissions from
multiple small fields located near each other using a combination of a dispersion model and a set of passive
diffusion sensors which integrate over a few hours to weekly periods. We found that the Philip (1959) analytical
model gave similar concentrations as the backward Lagrangian Stochastic model WindTrax (using the Monin
and Obukhov parameterisation) above a small source, as long as the stability correction functions used in both
models are similar.
We demonstrated by theoretical considerations that passive sensors always lead to the underestimation of
ammonia emissions for an isolated source because of the negative time correlation between the ammonia
emissions and the transfer function. Using a yearly meteorological dataset typical of the oceanic climate of
western Europe we found that the bias over weekly integration times is typically -8±6%, which is in line with
previous reports for bird colonies. Larger biases are expected for meteorological conditions with stable
conditions and low wind speeds typical of continental climates, as soon as the integration period is larger than 12
hours.
We showed that the quality of the inference method for multiple sources was dependent on the number of
sensors considered above each plot. The most essential technique to minimise the bias of the method was to
place a sensor in the middle of each source within the boundary layer. The quality of the sensor positioning was
evaluated using "condition numbers" which ranged from 2 to 3 for a sensor placed at 25 cm above the ground to





much higher values (40-1.6×10$^5$) for a sensor at 2 m height above 25 m width sources. Although the highest
sensors had low condition numbers, they were shown to improve the robustness of the sources inference
especially for evaluating the background concentrations. Using replicates of each treatment was found to be
essential for evaluating the quality of the inference and derive robust statistical indicators for each treatment.

When considering a system, characteristic of agronomic trials, composed of a low and a high potential source
and a reference with no nitrogen application, we found that the fractional bias remained smaller than around 25%
for ratios between the largest to the smallest sources lower than factor 5 and increased as a power function of the
ratio. Furthermore, the dynamics of the emissions were found not to strongly affect the fractional bias. As
expected, we also found that the fractional bias decreased with increasing source dimensions, especially for the
lowest source strength in a multiple source trial.

Finally, a test on a practical trial proved the applicability of the method in real situations with contrasted
emissions. We indeed calculated ammonia emissions of around $10.1 \pm 0.7\%$ of the total ammoniacal nitrogen
applied for surface applied slurry while we found less than 1% emissions for the treatments with incorporated
slurry.

This method could also be improved by incorporating knowledge of the surface source dynamics into the
inference procedure. Further work is required however, for validating the method, for instance using prescribed
emissions, and to evaluate it for growing crops using real measurements with diffusion samplers close to the
ground.

**Acknowledgements**

This study was supported by EU FP7 NitroEurope-IP (grant number 017841) and ECLAIRE (grant number
282910), French national projects CASDAR VOLAT'NH3 (grant number 0933), ADEME EVAPRO (grant
number 1560C0036), ADEME EVAMIN (grant number 1660C0012). The data sets used in this paper can be
obtained from the authors upon request. The meteorological dataset used in this study are from the ICOS site FR-
GRI which can be obtained from http://fluxnet.fluxdata.org/. We thank Erwan Personne for the use of the
Surfatm model.

**Supplementary material**

(see supplementary material manuscript)

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
