# Peer review of "Evaluation of a new inference method for estimating ammonia"

_Biogeosciences, 2017_

## Referee Comment (RC1) · Anonymous Referee #1 · 6 Dec 2017

This study carried out are providing answers to the much discussed question about the effect of having many plots in the field on measured ammonia emission from manure applied on the plots. Exploring the effect of measuring average ammonia concentration for increasing time intervals, the numbers of measuring heights and the best heights for measuring the emission. The answers to these questions are most important and the issue is discussed by scientist in Europe especially after the publications of Sinterman et al. questioned the existing design of measuring ammonia emission.

The authors have developed a model for calculating emission of ammonia from as it varies over the day and year as affected by surface soil temperature, wind and atmospheric stability. Then, as I understand the paper, they calculate how much the emitted ammonia will contribute to atmospheric ammonia concentration at different heights above the soil surface as it is affected by climate and plot size i.e. the loss pattern over time after volatilization start is assessed using decay curves of source strength.

The atmospheric NH3 concentration data, climate data are then used as input to model calculation of the emission from a plot and plots in a field as affected a range of different management of measuring ammonia concentration, height of the ammonia conc. Measurments,.number of plots affecting ammonia concentration in plots downwind a plot, plot size etc.

This reviewer is not a specialist in micrometeorology so I can not evaluate the quality of the model calculations. In the following is my impression of the presentation and interpretation of the data.

Abstract

Line 9 NH3 is presented but later the authors write ammonia – should be NH3

Line 10: the abbreviation N for nitrogen should be given and N used in the text.

I am not familiar with the term inference method, the term inferring, inferred in this context? May be because my native language is not English.

L68: What is an intensive source?

L69-70: require hourly concentrations of what????

L87 Multiple-source inversion problem?

L121-124: Units are missing

L324-327: Has the data from this experiment been used in previous articles, reports, procedings?

L355-356: Rewrite

383: Condition number – what is this – referring to an equation S1 in annex, it is a number often used so a presentation of how it is calculated should be given in the article

P519: When discussing the effect of height for measuring the horizontal then the authors should relate the outcome of their study to that of Wilson et al. who showed on baisis of micro-met. Calculations that there is a best height for measuring the horizontal flux at one height (This Zinst height is higher that the height recommended here)

L555: What is the highest source?

Figures: The font size of the Y and X axis and some of the legends are too small on most figures. On some figures there are too many lines (7 lines on fig 4) making it very difficult to see the individual lines.

Figure 5 & 8: I assume that prescribed is the emission data provided by calculation and inferred is emission calculated by knowing NH3 conc. At 0.5 m and weather conditions.

Fig. 7: Need improvement

Fig 9; Why not mention the emission strength of the source instead of Treatment 1-3, áťę = 104 (what is the units?)

Fig 16: Is it correct that measured emissions are not included – if so then the measured results should be included?

---

## Short Comment (SC1) · 20 Dec 2017

**Comment to the paper „Evaluation of a new inference method for estimating ammonia volatilisation from multiple agronomic plots"by Benjamin Loubet et al.**

*Albrecht Neftel*
*Neftel Research Expertise, CH-3033 Wohlen b. Bern, CH, Switzerland*

*Christoph Häni*
*School of Agricultural, Forest and Food Sciences, Bern University of Applied Sciences, CH-3052 Zollikofen, CH, Switzerland*

This paper presents an appealing and low-cost approach to determine $NH_3$ losses from adjacent multiple agronomical plots by a combination of concentration measurements using passive sampler devices and a dispersion model that is driven by turbulent parameters inferred from standard 30 minutes meteorological data.

The aim of the paper is described as: *"Can inverse dispersion modelling approaches be used for inferring $NH_3$ emissions from multiple small plots (agronomic trials) using passive samplers, and to which degree of accuracy?"*

The overall answer is encouraging with the statement in the conclusions *"In this study we have demonstrated that it is possible to infer with reasonable biases ammonia emissions from multiple small fields located near each other using a combination of a dispersion model and a set of passive diffusion sensors which integrate over a few hours to weekly periods"*.

According to our judgement the accuracy will be mainly determined by two aspects addressed below.

a) Bias related to the applied dispersion modelling

Dispersion models are a simplified mathematical representation of the turbulent motion in the surface layer and will always deviate from reality. Systematic biases can be expected when modelling the lower heights of the measurements that are discussed in this paper. For concentration sensors place close to the ground (e.g. 25cm above ground) transfer functions are likely to be biased due to e.g. the needed simplifications that must be made to describe the exchange process at the ground, the natural heterogeneity on a small scale at the surface or the violation of model assumptions such as the failure of K-theory close to the canopy (Raupach and Legg, 1984). Furthermore, the translation of the sensor height in a model framework is challenging for very low heights since the sensor height value (and with that the resulting value of D at that location) becomes very sensitive to sensor height measurement errors as well as to the absolute values of z0 and d. To us, a sensor height of 25 cm seems too close to the surfaces.

The authors are using their FIDES-3D model that is based on an analytical solution of the advection-diffusion equation. This model is compared with the backward Lagrangian Stochastic dispersion (bLS) model described in Flesch et al. (2004) (the "WindTrax" software, Thunder Beach Scientific, Nanaimo, Canada). For the presented analysis the FIDES model $K_z$ was adopted to match the far field approximation of $K_z$ of the bLS model. We are missing an explanation, why this was done.

In the supplement, a detailed investigation is presented how the two models differ in their formulation of the vertical diffusivity $K_z$. The assumed far field vertical diffusivity in the bLS model

is approximated by parametrizations provided in Flesch et al. (1995). We would like to remark that WindTrax uses slightly different default parametrizations of $\sigma_w$ than provided in Flesch et al. (1995) (see e.g. the manual on the WindTrax homepage[1]). This is resulting in vertical diffusivities given as:

$$K_z(Z) = 0.5 * 1.25 * u_* Z / (1 + 5Z/L) \qquad \text{for z/L} \geq 0$$

$$K_z(Z) = 0.5 * 1.25 \\ * u_* Z \times (1 - 6Z/L)^{0.25}(1 - 3Z/L)^{(1/3)} \qquad \text{for z/L} < 0$$

with a Schmidt number value of $Sc \cong 0.64$ for near-neutral stabilities with a smooth transition from L = ∞ to L = -∞. These equations differ from the equations S7 and S8, and imply a different interpretation of the differences between FIDES and WindTrax, though without changing the numeric results of the comparison.

In the supplement Figure S3 presents a comparison between evaluated concentrations with FIDES and WindTrax respectively using the prescribed emission sources with the SVAT model. This figure is hiding the apparent differences as a double logarithmic representation is used and the concentrations are shown using an emission source that shows a positive correlation between the meteo input parameters of the models and the source strength.  E.g. for neutral conditions the regression of ratio of the concentrations calculated with FIDES and WindTrax  at a height of 0.25m is indicated as  $c_{FIDES} = 0.97 \cdot c_{Windtrax}{}^{0.87}$.   For a concentration of 1 the ratio is 0.97 and for a concentration of 100 the ratio becomes 0.53. As the transfer function D in FIDES and WindTrax are only depending on the prevailing turbulent parameters it would be more instructive to use a constant unit emission of 1 and show the ratio on a linear scale as function of u* and L in a similar way as the authors have done in a previous paper (Carozzi et al., 2013).

b)   Bias related to the concentration measurements

The use of passive diffusive samplers is a challenging business. Within different networks the reliability of PS such as ALPHA samplers or Radiellos have been proven, but the use of them close to emitting source showed major deviations compared to other measurements. E.g. Misselbrook et al. (2005) found severe overestimations of passive diffusive samplers. The latest investigation stems from the Dronten experiment and is discussed in a paper by Michael Bell et al. (submitted to AFM). In this experiment the ALPHA samplers were affected by a positive bias in the order of 50% relative to the other devices.  We speculated that the exposure of the PS with the protection hat above them cached eddies from below loaded with higher $NH_3$ concentrations but shielded eddies with lower concentrations from above.

Figure 1 illustrates the $NH_3$ dynamic that occur over an emitting surface. The concentration was measured with a fast device described in Sintermann et al. (2011). Immediately after application of slurry with a splash plate the $NH_3$ concentration was measured at a height of 1m above ground with an ionization technique and a strongly heated inlet line to avoid as much as possible damping effects.
* * *
[1] http://www.thunderbeachscientific.com/downloads/atmosphericdata.pdf

[Figure]

Figure 1: NH$_3$ concentration timeseries measured 1m above ground over a manured surface with splash plate.

Concluding comments:

We judge that the most important potential biases of the proposed multiplot approach are related to biases of the concentration measurements and the used dispersion coefficient.  It would be instructive to calculate probability density functions of the estimated emissions with a dataset that reflect the distributions of the measurements and the turbulence parameters that drive the dispersion model.

The authors have tested their setup in field trial in April 2011 applying slurry with a DM content of 6% and an application rate of 41 kg N-NH$_3$/ha. According to the details given in the text, we assume that broadband application was used and was compared to fast incorporation and no application. The cumulated loss amounted to 8 to 10% of the applied NH$_3$. For broadband application, this is a loss on the low side (see e.g. Häni et al.,2016). We would not be astonished if the real emissions would be double as high.

The presented approach to perform NH$_3$ emission measurements in a multiplot arrangement is encouraging and goes in a good direction. To make the approach more robust, the employed ALPHA NH$_3$ sampling systems should be validated under real conditions, i.e. over an emitting source in comparison with e.g. MiniDOAS systems (Sintermann et al., 2016).

Finally, we would like to invite the authors to collaborate with us to compare the FIDES and WindTrax approach. We have an extensive dataset from field trials where we released CH$_4$ or a mixture of NH$_3$ and CH$_4$ from a circular artificial source with a diameter of 20 meters (Häni et al., 2017).

References:

M.W. Bell, A. Hensen, A. Neftel, B. Loubet , P. Robin , Y. Fauvel , Y. Hamon ,   M. Haaima, A.J.C. Berkhout , D.P.J. Swart, W.C.M. van den Bulk, B.F. van Egmond, D. van Dinther, A. Frumau, B. Esnault, C. Decuq, C.R. Flechard  Quantifying ammonia emissions from plot-scale and farm-scale sources using integrated mobile measurements and inverse dispersion modelling, (submitted to AFM).

Carozzi, M., Loubet, B., Acutis, M., Rana, G., Ferrara, R.M., 2013. Inverse dispersion modelling highlights the efficiency of slurry injection to reduce ammonia losses by agriculture in the Po Valley (Italy). Agric. For. Meteorol. 171, 306–318. 10.1016/j.agrformet.2012.12.012.

Flesch, T.K., Wilson, J.D., Harper, L.A., Crenna, B.P., Sharpe, R.R., 2004. Deducing ground-to-air emissions from observed trace gas concentrations: A field trial. J. Appl. Meteorol. 43 (3), 487–502.

Flesch, T.K., Wilson, J.D., Yee, E., 1995. Backward-Time Lagrangian Stochastic Dispersion Models and Their Application to Estimate Gaseous Emissions. J. Appl. Meteorol. 34 (6), 1320–1332.

Häni, C., Sintermann, J., Kupper, T., Jocher, M. and Neftel, A., 2016. Ammonia emission after slurry application to grassland in Switzerland. Atmospheric Environment, 125: 92-99.

Häni, C., Voglmeier, K., Jocher, M., Ammann C., 2017. Recovery rates from line-integrated NH3 and CH4 measurements using backward Lagrangian stocahstic dispersion modelling. Geophysical Research Abstracts, Vol. 19, EGU2017-19557, 2017

Misselbrook, T.H, Nicholson, F.A., Chambers, B.J.,Johnson, R.A. ,2005, Measuring ammonia emissions from land applied manure: an intercomparison of commonly used samplers and techniques. Environmental Pollution 135 (2005) 389–397

Raupach, M.R., Legg, B.J., 1984. The uses and limitations of flux-gradient relationships in micrometeorology. Agricultural Water Management 8 (1-3), 119–131. 10.1016/0378-3774(84)90049-0.

Sintermann, J., Spirig, C., Jordan, A., Kuhn, U., Ammann, C., and Neftel, A.: Eddy covariance flux measurements of ammonia by high temperature chemical ionisation mass spectrometry, Atmos. Meas. Tech., 4, 599–616, doi:10.5194/amt-4-599-2011, 2011.

Sintermann, J., Dietrich, K., Häni, C., Bell, M., Jocher, M., Neftel, A., 2016. A miniDOAS instrument optimised for ammonia field measurements. Atmos. Meas. Tech. 9 (6), 2721–2734. 10.5194/amt-9-2721-2016.

---

## Referee Comment (RC2) · Anonymous Referee #2 · 29 Jan 2018

Loubet and others study a new method for inferring ammonia loss from small agricultural plots.

I found the modeling analysis to make for an interesting case study regarding the applicability of field experiments. The approach treats bias errors carefully. As a consequence, I feel that the manuscript makes an earnest effort to quantify biases associated with passive ammonia sampling over small agronomic field plots and will be a valuable contribution to the literature.

Minor comments: 'Further work should anyway be produced for validating this method in real conditions' at the end of the abstract does not sound hopeful. Rather, the

authors should try to discuss strategies for further improving the method and reducing uncertainties.

Line 41: 55.3% sounds remarkably specific given uncertainties in measuring NH3 flux.

53: 'most of the time large fields' is awkward wording.

57: agronomic trials are not necessarily of those dimensions.

Parentheses on line 67.

118: quotes are unnecessary.

On 130, what is the typical reaction time (and thereby Damkoehler number?)

I find the tau near the overbar in 2 and other equations to be a bit distracting because it could be confused with an exponential term.

Equation 4 could be rearranged to reflect that only the numerator of the second term on the right hand side is unknown.

251: why is zref 3.17 m? The curly braces in Rb{NH3} I find to be a bit distracting.

263: is there a justification for the model in simulation 2?

265: what are typical parameters for the Gaussian model? Also, what mechanism causes it? The urea spreader?

267: I understand why 4.6 now in simulation 2. . .but why does this 'best' represent NH3 emissions?

302: why is the covariance term negligible at the half hourly period? The spectral gap in eddy covariance studies?

303: in 2.5.3, these are not hypotheses as they cannot be falsified, even in the model.

327: extra period

336: how close is 'nearby'? From the figure it looks like it was part of the larger setup.

355: results should be written in the past tense.

365: define Gamma for the reader in the figure legend.

Please avoid using red and green simultaneously in Figure 4. This figure appears to be made using R, and gray is also a default color. And honestly yellow is never a good choice on a white background.

384: focuses

Figure 6 confuses me a bit because the 13 periods vary so strongly in their meteorological conditions from summer to winter, why are they grouped? The bars also leave the figure in the upper left subplot.

464-466: the attribution of stability with respect to continental vs. oceanic sites is too much of an approximation. There are many continental sites that are consistently windy, often due to orography.

There is a strange x on line 468.

Font sizes for figure 7 should be increased.

741: why bird colonies?

742: again, continential does not imply low wind speeds.

---

## Author Response (AR1)

[revised manuscript text omitted]

$$K_z(Z) = 0.5 \times 1.25 \times u_*Z / \left(1 + 5\frac{Z}{L}\right) \qquad \textbf{for L > 0} \qquad \text{(S9)}$$

$$K_z(Z) = 0.5 \times 1.25 \times u_*Z \times \left(1 - 6\frac{Z}{L}\right)^{0.25} \left(1 - 3\frac{Z}{L}\right)^{1/3} \qquad \textbf{for L} \leq \textbf{0} \qquad \text{(S10)}$$

**From the Eqns. (S9 and S10) is noticeable that under near neutral situations ($L \to +\infty$ or $L \to -\infty$), the diffusivity $K_z(Z)$ is converges to $ku_*Z \times 0.64^{-1}$ (where $k = 0.41$ is the von Karman's constant and 0.64 represents the Schmidt number) and is continuous for all $L$. Figure S1 based on Eqns. (S9 and S10)** shows that our approach insures a coherency between the diffusivity of the bLS and Philip approach but small differences remain which are height dependent. We should also notice that lateral dispersion was treated separately in the two models, which will also lead to differences in the modelled concentration, especially for larger fields.

[Figure]

**Figure S1. Ratio of the "tuned" FIDES ("Philip") to WindTrax vertical diffusivity for scalars ($K_z(z)$) as a function of the inverse of Obukhov length ($1/L$) at 0.25, 0.5, 1 and 2 m heights. The tuned diffusivity corresponds to Eqns. (S9 and S10).**

**S4.3. Comparison of FIDES and WindTrax models for predicting concentrations above a single source**

A first step in the study was to compare the two dispersion models. **Figure S2** shows that the "tuned" FIDES

model leads to the same concentration pattern as WindTrax **at 0.5 m above the source,** although systematically underestimating the maximum concentration under unstable conditions. **This behaviour is clearly visible in**

**Figure S3** **where the concentrations modelled with the "tuned" FIDES at 2 m above the surface (right**

**graph) are underrated by about 15% for unstable conditions compared to WindTrax, while matching for**

**stable and neutral conditions. We** further see that the concentration modelled with the original FIDES (Philip,

1959) are similar at 25 cm above the surface (left graphs) but differ substantially at 2 m above the surface (right graphs). This is expected as the longer the travel distance, the larger the expected difference in dilution if the two models' diffusivity differ. In the original FIDES, the diffusivity is lower than in WindTrax by a factor of roughly

**one and half** ($Sc^{\text{Philip}} = 1$ and $Sc^{\text{WT}} = 0.$**64**). In a first order approach (over an infinitely homogeneous source), the concentration difference between $z_0$ and 2 m would be proportional to the aerodynamic resistance (itself proportional to the inverse of the vertical diffusivity) times the height above ground (see e.g. Flechard et al.,

2013), which explains the differences observed in **Figure S3**.

[Figure]

**Figure S2. Example concentration modelled above a single ammonia source using two dispersion models WindTrax**
**and FIDES with $K_z$ as in Philip (1959), at 0.5 m above a simulated squared ammonia source of 25 by 25 m in the FR-**
**Gri ICOS site during August 2008.**

[Figure]

**Figure S3. FIDES versus WindTrax concentration modelled above an ammonia source of 25 × 25 m at 0.25 and 2 m heights. In these graphs the FIDES vertical diffusivity $K_z$ is either as in Philip (1959) (b) or fitted to Flesch et al. (1995) (a) and (c) as explained in S4.2. The comparison is made over the entire year of 2008 in the FR-Gri ICOS site. S, U and N stand for stable, unstable and neutral atmospheric conditions. The linear regression equation is given for each condition together with the $R^2$ of that regression. The black line is the 1:1 line. (a) and (b) show log-log axes an power law fits while (c) shows linear axes and linear fits.**

**Figure S3** also shows that the "tuned" FIDES modelled concentrations (top graphs) do not perfectly fit to the **WindTrax** ones (top graphs in **Figure S3**). At height of 25 cm, the "tuned" FIDES concentration does lead to a worse regression score than the original FIDES. Although **Figure S3** is focussing on a 25 m × 25 m field, the results are similar for larger fields (data not shown). This is explained by the difference in $Z$-dependency of $K_z$ in the WindTrax and FIDES model, which is highlighted in **Figure S1**: under stable conditions ($1/L > 0$), "tuned" FIDES $K_z(Z)$ is larger than WindTrax at 0.25 and 2 m, but smaller at 0.5 and 1 m, and the opposite under unstable conditions ($1/L < 0$). This means that constitutively the two models may never fit perfectly, showing a bias that will depend on height. Nevertheless, the correlation between the two models is very high as shown by large $R^2 \geq \sim 0.96$.

[Figure]

Figure S4. Relative difference between FIDES and WindTrax concentrations as a function of the stability parameter (1/L). Data refer to the same conditions reported in Figure S3.

Figure S4 reports the relative difference between the $NH_3$ concentration calculated by the two models at 0.5 and 2 m height, as a function of the stability parameter (*1/L*). As previously stated, under unstable conditions FIDES clearly underestimates the concentrations up to 30% at 0.5 m and lower heights, while this gap is reduced and more scattered at 2 m height. Moving towards neutral conditions the two models tends to agree notwithstanding an overestimation to 10% by WindTrax at 2 m height concurrently with an underestimation of the same magnitude by FIDES at 0.5 m. Under stable conditions there is a clear agreement at 2 m height, while this correspondence remains unbalanced to lower heights.

**Supplementary figures**

[Figure]

Figure S5. (a) Distribution of condition numbers for the 0.25 m height sensor and the 25 m width plots, for integration periods of 6h and 24h, and (b) condition number as a function of 1/*L*, where *L* is the Obukhov length.

**References quoted in the supplementary material**

Carozzi, M., Loubet, B., Acutis, M., Rana, G. and Ferrara, R.M., 2013. Inverse dispersion modelling highlights
the efficiency of slurry injection to reduce ammonia losses by agriculture in the Po Valley (Italy).
Agric. For. Meteorol., 171: 306-318.
Crenna, B.R., Flesch, T.K. and Wilson, J.D., 2008. Influence of source-sensor geometry on multi-source
emission rate estimates. Atmos. Environ., 42(32): 7373-7383.
Flechard, C.R. et al., 2013. Advances in understanding, models and parameterizations of biosphere-atmosphere
ammonia exchange. Biogeosciences, 10(7): 5183-5225.
Flesch, T.K., Harper, L.A., Desjardins, R.L., Gao, Z.L. and Crenna, B.P., 2009. Multi-Source Emission
Determination Using an Inverse-Dispersion Technique. Boundary-Layer Meteorology, 132(1): 11-30.
Flesch, T.K., Wilson, J.D., Harper, L.A., Crenna, B.P. and Sharpe, R.R., 2004. Deducing ground-to-air
emissions from observed trace gas concentrations: A field trial. J. Appl. Meteorol., 43(3): 487-502.
Flesch, T.K., Wilson, J.D. and Yee, E., 1995. Backward-Time Lagrangian Stochastic Dispersion Models and
Their Application to Estimate Gaseous Emissions. J. Appl. Meteorol., 34(6): 1320-1332.
Gao, Z.L., Desjardins, R.L., van Haarlem, R.P. and Flesch, T.K., 2008. Estimating Gas Emissions from Multiple
Sources Using a Backward Lagrangian Stochastic Model. Journal of the Air & Waste Management
Association, 58(11): 1415-1421.
Kaimal, J.C. and Finnigan, J.J., 1994. Atmospheric Boundary Layer Flows, Their structure and measurement.
Oxford University Press., New York, 289 pp.
Kormann, R. and Meixner, F.X., 2001. An analytical footprint model for non-neutral stratification. Boundary
Layer Meteorol., 99(2): 207-224.
Loubet, B., Milford, C., Sutton, M.A. and Cellier, P., 2001. Investigation of the interaction between sources and
sinks of atmospheric ammonia in an upland landscape using a simplified dispersion-exchange model. J.
Geophys. Res.-Atmos., 106(D20): 24183-24195.
Philip, J.R., 1959. The Theory of Local Advection .1. J Meteorol, 16(5): 535-547.
Sutton, O.G., 1932. A Theory of Eddy Diffusion in the Atmosphere. Proceedings of the Royal Society of
London. Series A, 135(826): 143-165.
Wilson, J.D., 2015. Computing the Flux Footprint. Boundary Layer Meteorol., 156(1): 1-14.

**bg-2017-424-RC1: answer to referee 1 comments.**

We would like to thank referee 1 for his helpful comments which we have answered below.

**General Comments**

This study carried out are providing answers to the much discussed question about the effect of having many plots in the field on measured ammonia emission from manure applied on the plots. Exploring the effect of measuring average ammonia concentration for increasing time intervals, the numbers of measuring heights and the best heights for measuring the emission. The answers to these questions are most important and the issue is discussed by scientist in Europe especially after the publications of Sinterman et al. questioned the existing design of measuring ammonia emission

The authors have developed a model for calculating emission of ammonia from as it varies over the day and year as affected by surface soil temperature, wind and atmospheric stability. Then, as I understand the paper, they calculate how much the emitted ammonia will contribute to atmospheric ammonia concentration at different heights above the soil surface as it is affected by climate and plot size i.e. the loss pattern over time after volatilization start is assessed using decay curves of source strength.

The atmospheric NH3 concentration data, climate data are then used as input to model calculation of the emission from a plot and plots in a field as affected a range of different management of measuring ammonia concentration, height of the ammonia concentration measurements, number of plots affecting ammonia concentration in plots downwind a plot, plot size, etc.

This reviewer is not a specialist in micrometeorology so I cannot evaluate the quality of the model calculations. In the following is my impression of the presentation and interpretation of the data.

**Abstract**

Line 9 NH3 is presented but later the authors write ammonia – should be NH3

This is a sound remark; we agree that we can use $NH_3$ throughout the manuscript once it has been defined as "tropospheric ammonia" except when it is the first word of a sentence.

Line 10: the abbreviation N for nitrogen should be given and N used in the text.

We thank the referee for the suggestion; we however think we could stick with "nitrogen" to avoid too many abbreviations in the abstract.

I am not familiar with the term inference method, the term inferring, inferred in this context? May be because my native language is not English.

We agree with the referee that "to infer" may not be a very commonly used term. It is a synonym of "to deduce". We hence speak about a "source inference method" in the sense that the method is used to "deduce" the ammonia source.

L68: What is an intensive source?

We thank the reviewer for spotting this term which we might have mis-used. We rather wanted to mean an intense or strong source. We propose to change term the "intensive" to "strong" in the manuscript.

L69-70: require hourly concentrations of what????

Of $NH_3$. We propose to add this precision in the text.

L87 Multiple-source inversion problem?

We thank the reviewer for spotting this incoherency. Actually, we defined what we meant in lines 76-77, as "the multiple source problem, which consists of inferring multiple sources based on measured concentrations at multiple points in space and time,…". We hence propose to use the term "multiple-source inference problem" throughout the manuscript to keep it coherent.

L121-124: Units are missing

Thanks for spotting that the units for concentration and emissions were missing. The concentration and source are in µg N-$NH_3$ m$^{-3}$ and µg N-$NH_3$ m$^{-2}$ s$^{-1}$, respectively. We propose to add these precision in the manuscript.

L324-327: Has the data from this experiment been used in previous articles, reports, proceedings?

Results from this experiment have been used jointly with other experimental trials in a poster proceeding, but not for testing the multiple-source inference methodology presented here. The poster was presented at a French meeting on fertilisation. The objective was to compare the emissions potential from several treatments based on the use of a "gradient" method applied to badges. The objective of the poster presentation was to show the potential of using alpha badges to differentiate nitrogen application methods in terms of potential ammonia losses. The link to the poster abstract in French is here:

http://www.comifer.asso.fr/images/pdf/11emes_rencontres/Interventions/Session%201/5%20-%20Jean-Pierre%20COHAN/Article%20Jean-Pierre%20COHAN.pdf.

L355-356: Rewrite

We propose to change to the following simplified sentence: "The friction velocity u* varied between 0.024 and 1.181 m s-1, and the stability parameter z/L varied between -49 and 21 m-1 (Figure 3)."

383: Condition number – what is this – referring to an equation S1 in annex, it is a number often used so a presentation of how it is calculated should be given in the article

The condition number is indeed an important indicator of the geometry of the multiple sources inference problem. We feel that it was well defined and discussed in the supplementary material section S2. The way it is calculated in practice is mentioned at line 46 of the supplementary material. We propose to slightly modify this sentence to make it even clearer: "In practice, the calculation of CN was performed using the kappa function in R (version 3.2.3)." We also propose to add the phrase "(see supplementary material section S2)" to lines 384 and 684 of the manuscript where this indicator is mentioned.

P519: When discussing the effect of height for measuring the horizontal then the authors should relate the outcome of their study to that of Wilson et al. who showed on basis of micro-met. Calculations that there is a best height for measuring the horizontal flux at one height (This Zinst height is higher that the height recommended here)

Although the ZINST method has no link with our approach, it is an interesting remark that brought back to our attention that the heights at which the alpha badges should be placed would depend on roughness length and displacement height. Indeed, the ZINST method is a method that uses the finding of Wilson et al. (1982) that the ratio of the source strength to the horizontal flux at height ZINST is somewhat constant whatever the stability conditions. Interestingly, Wilson et al. showed that ZINST was an exponential function of $z_0$ for a given source diameter. We hence propose to add the following text after Line 519: "It is interesting to note that the height which was found to provide an optimal inference of $NH_3$ sources (below 0.5 m) is smaller than the ZINST reported by Wilson et al. (1982) (which was 0.9 m for 40 m diameter circular sources, and which we estimate as 0.65 m based on a power law extrapolation as in Laubach et al. (2012)). It is also important to note that this height should vary with both the roughness length z0 and displacement height as was showed by Wilson et al. (1982) for ZINST."

L555: What is the highest source?

We actually meant the largest source. We propose to change the text in L555 and L540 and also Figure 12 accordingly.

**Figures:**

The font size of the Y and X axis and some of the legends are too small on most figures. On some figures there are too many lines (7 lines on fig 4) making it very difficult to see the individual lines.

We thank the reviewer for his suggestion. We have looked at the figures thoroughly again and we agree indeed that some figures may be difficult to read, but most figures look good to us. We propose to improve some figures as explained below but we would like to have the editor's point of view for the other figures.

- Figure 4: we propose to reduce the number of integration periods and to keep only 0.5h, 24h and 168h which are sufficient to show the variability that is lost by integrating concentration measurements.

[Figure]

Figure 4. Example modelled concentration pattern at 1 m above a single 50 m width source for several averaging periods (0.5h, 12h and 168h) for the month of July 2008. The source Γ was set to $10^5$. The y-axis is log scaled.

- Figures 5: there were some legends that we left in the right corner but these are not useful as the main legend was written at the top of the graph. We propose to erase these legends:

[Figure]

**Figure 5. Example source inference for a 25 m width square field and a concentration sensor placed at 0.5 m above ground. Here Γ = 10000 and is set to constant (pattern 1). The 7 integration periods are shown: 0.5h to 168h. The x-axis shows the day of year and corresponds to a span over November. The prescribed source is in black (Obs.) and the inferred one in red (Pred.)**

- Figure 7: we propose to change the font size and the Y axis label. We will also change the caption for $u_*$ and $1/L$ classes:

[Figure]

**Figure 7. Relative root mean squared error as a function of integration period for stability factor and friction velocity classes for a single 25 m side field. Medians and quartiles are given for equally sized bins of $u_*$ and $1/L$ and for the lowest sensor height (0.25 m). The blue, pink and green curves are the 3rd, 2nd and 1st quartiles, respectively.**

Figure 5 & 8: I assume that prescribed is the emission data provided by calculation and inferred is emission calculated by knowing NH3 concentration at 0.5 m and weather conditions.

Indeed prescribed emissions are calculated using equations (9) and (10) with a constant Gamma, and inferred are those based on measured concentration at 0.5 m height and transfer coefficient using equation (7).

Fig. 7: Need improvement

We agree. See previous section for our proposition.

Fig 9; Why not mention the emission strength of the source instead of Treatment 1-3 (what is the units?)

This is indeed a sound remark. We propose indeed to use the emission potential $\Gamma$, which actually has no units. Figure 9 would look like this (Figures 10-12 would be changed accordingly):

[Figure]

**Figure 9. Effect of integration period on source inference in the multiple-plot setup. The fractional mean bias of the source is shown for each treatment. Inference strategy C1 was used (single sensor, independent blocks, and background concentration known). Statistics for runs with target heights 0.25 and 0.5 m and source side = 25 m are calculated. All application periods are considered. Filled points show medians, boxes show interquartiles and bars show minimums and maximums. Outliers are points to 1.5 times away from boxes limits.**

Fig 16: Is it correct that measured emissions are not included – if so then the measured results should be included?

There might be a misunderstanding of that figure; indeed Figure 16 reports inferred emissions using the multiple-source inference method that was presented and discussed in this manuscript. But in that experiment no other method than this one was used to "measure" emissions. This is actually our point to show that this inference method is appropriate for estimating the emissions under real situations. In a way our inference method gives a measurement of the ammonia emissions.

**References quoted in the answer to reviewer 1**

Laubach, J., Taghizadeh-Toosi, A., Sherlock, R.R. and Kelliher, F.M., 2012. Measuring and modelling ammonia emissions from a regular pattern of cattle urine patches. Agric. For. Meteorol., 156: 1-17.

Wilson, J.D., Thurtell, G.W., Kidd, G.E. and Beauchamp, E.G., 1982. Estimation of the rate of gaseous mass transfer from a surface source plot to the atmosphere. Atmospheric Environment (1967), 16(8): 1861-1867.

**Answer to Referee 2 comments**

We thank Referee 2 comments which we hope will help us improving our manuscript.

**General Comments**

Loubet and others study a new method for inferring ammonia loss from small agricultural plots. I found the modelling analysis to make for an interesting case study regarding the applicability of field experiments. The approach treats bias errors carefully. As a consequence, I feel that the manuscript makes an earnest effort to quantify biases associated with passive ammonia sampling over small agronomic field plots and will be a valuable contribution to the literature.

**Minor comments**:

'Further work should anyway be produced for validating this method in real conditions' at the end of the abstract does not sound hopeful. Rather, the authors should try to discuss strategies for further improving the method and reducing uncertainties.

We thank the reviewer for this comment. We have indeed identified two strategies for further improving the method: (1) using Bayesian inference which has the potential for constraining the emissions and avoiding unrealistic sources inference as shown by Yee and Flesch (2010), and (2) changing the cost function (also called objective function); instead of inferring the emission strength, we could infer the emission potential $\Gamma$ (a strictly positive number). This last method has the advantage of avoiding non-plausible deposition fluxes, because the flux is calculated as the concentration above the source minus the concentration at the ground, divided by transfer resistance. With such an approach negative fluxes (deposition) can occur within the limit of plausible transfer resistances but not above. We believe that the combination of these two strategies has the potential to improve the method substantially.

We think that "calibration" of the method against controlled sources is a remaining challenge that needs to be tackled (as also suggested by the comments of A. Neftel and C. Hanni in the interactive discussion).

We hence propose to add this more positive statement at the end of the abstract: "We believe that the method could be further improved by using Bayesian inference and inferring surface concentrations rather than surface fluxes. Validating against controlled sources is also a remaining challenge."

Line 41: 55.3% sounds remarkably specific given uncertainties in measuring NH3 flux.

This is a sound comment. We propose to change to 55%.

53: 'most of the time large fields' is awkward wording.

Indeed. We propose to change to "most of the time also requires the use of large fields"

57: agronomic trials are not necessarily of those dimensions.

This is a sound remark indeed and we should not be as general as we were. We would propose to change to: "Especially useful for measuring ammonia losses are methods that can deal with small and medium-scale fields (20-50 m on the side) that are commonly used in agronomic trials."

Parentheses on line 67.

Thanks for spotting this. We have withdrawn the left parenthesis.

118: quotes are unnecessary.

We agree and have withdrawn them.

On 130, what is the typical reaction time (and thereby Damkoehler number?)

Typical Damköhler numbers showed by Nemitz et al. (2009) above a cut grassland canopy fertilised with ammonium nitrate were from 0.001 to 1. Values greater than 0.1 only occurred marginally, and usually during night-time conditions (Figure 6 in Nemitz et al. 2009). We would of course expect larger Damköhler number values for slurry application which may generate larger concentrations than those reported by Nemitz et al. (2009), or with surface canopies having larger residence times. But in any case we expect the chemical depletion of ammonia to remain small at the spatial scale we are focussing on (around 200-300 m).

I find the tau near the overbar in 2 and other equations to be a bit distracting because it could be confused with an exponential term.

This is a sound remark. We propose to remove the *taus* and just leave an explanation in the text that the overbars denote averages over the period tau.

Equation 4 could be rearranged to reflect that only the numerator of the second term on the right hand side is unknown.

It is true that the numerator of the second term is the only unknown. However we can't see how to isolate this term apart from multiplying by D(x). Moreover, leaving the equation as it is now has the advantage of explicitly showing this term which is the bias. We hence propose to keep equation (4) as it is.

251: why is $z_{ref}$ 3.17 m? The curly braces in $R_b\{NH_3\}$ I find to be a bit distracting.

Regarding $z_{ref}$, we subjectively choose to use the reference height $z_{ref}$ as the height where our ultrasonic anemometer was placed in the field, which simplified the calculation of the aerodynamic resistance for us. This does not have much importance anyway as we assumed that atmospheric ammonia concentration was zero.

Regarding Rb, we propose to change Rb{NH3} to $R_{bNH_3}$.

263: is there a justification for the model in simulation 2?

Exponential decrease in emission potential is representative of strong $NH_3$ emissions like those happening following slurry application. The value of 4.6 and the time scale $\tau_0$ were chosen arbitrarily and would represent emissions a little bit less intense than those for nitrogen applications reviewed by Massad et al. (2010). In fact the equation we used here would be equivalent to a time scale equal to 6 days while in Massad et al. (2010) they report a time scale of 2.88 as being representative of slurry application. We propose to add the following text in Line 270: "The time scale of the exponential decrease we used here was around 6 days, which is twice as large as the one reported by Massad et al. (2010) for slurry application (2.9 days)."

265: what are typical parameters for the Gaussian model? Also, what mechanism causes it? The urea spreader?

The Gaussian model is rather representative of urea application. Indeed, $NH_3$ emissions result from combined processes: first the urea is hydrolysed by urease enzymes which release ammonium which can be volatilised but can also be nitrified or absorbed by roots. This leads to typical emissions starting a few days following application and showing a maximum up to 15 days following application but also a slower decrease of the emissions following the peak.

The Gaussian model was centred on day 14 with a standard deviation of 8.4 days.

267: I understand why 4.6 now in simulation 2. . .but why does this 'best' represent NH3 emissions?

As explained in previous paragraphs and following Massad et al. (2010) this model best represents slurry applications.

302: why is the covariance term negligible at the half hourly period? The spectral gap in eddy covariance studies?

The covariance term is indeed negligible at that time scale because of the so-called spectral gap in eddy covariance studies. This gap corresponds to time scales at which there is little energy in the turbulence and surface flux spectra (see e.g. Van der Hoven {, 1957 #25437}). We propose to replace the sentence at line 302 by "In practice the concentrations were computed at each sensor location using Eq. (6) over 0.5h: at that time scale, which corresponds to the spectral-gap, the covariance term is assumed to be negligible (Van der Hoven, 1957)."

303: in 2.5.3, these are not hypotheses as they cannot be falsified, even in the model.

This is indeed an interesting remark. We propose to change to the term "scenario" instead.

327: extra point

Thanks for spotting this. We have removed it.

336: how close is 'nearby'? From the figure it looks like it was part of the larger setup.

The meteorological data were measured at around 25 m away from the edge of the central plots (Figure 2). We propose to change the sentence for clarification: "The meteorological data were measured at less than 50 m from the central plots (Figure 2)".

355: results should be written in the past tense.

Thanks for the comment. We propose to change this sentence also to clarify its meaning : "The friction velocity u* varied between 0.024 and 1.181 m s-1, and the stability parameter z/L varied between -49 and 21 m-1 (Figure 3)"

365: define Gamma for the reader in the figure legend.

Thanks for the comment. We propose to change the last part of the legend to "…with an emission potential ⬚ = 10000"

Please avoid using red and green simultaneously in Figure 4. This figure appears to be made using R, and gray is also a default color. And honestly yellow is never a good choice on a white background.

The comment that Figure 4 was hard to read was also made by reviewer 1. We have hence simplified the Figure and we further propose to change the colors as suggested by reviewer 2:

[Figure]

Figure 4. Example modelled concentration pattern at 1 m above a single 50 m width source for several averaging periods (0.5h, 12h and 168h) for the month of July 2008. The source $\Gamma$ was set to $10^5$. The y-axis is log scaled.

384: focuses

Thanks for spotting this typo. We have corrected it.

Figure 6 confuses me a bit because the 13 periods vary so strongly in their meteorological conditions from summer to winter, why are they grouped? The bars also leave the figure in the upper left subplot.

The idea for grouping the periods in Figure 6 but also in Figures 9-11 and 14 is actually to evaluate the variability of the bias due to meteorological conditions: ideally, if the method shows little variability in the bias, this bias could be characterised and even withdrawn. In Figure 6 we try to give a broad view of how the bias changes with sensor height and plot width. Figure 7 actually shows the variability of the bias due to meteorological conditions.

Regarding the scale, we chose to have a single scale for all panels to ease the comparison between heights and plot size, and we also chose to get the scale focussed enough to better see biases in the range -0.2 to 0.1. What we conclude from the upper left subplot is that the bias is much larger than all other cases which shows that that combination height-plot size is not satisfactory.

464-466: the attribution of stability with respect to continental vs. oceanic sites is too much of an approximation. There are many continental sites that are consistently windy, often due to orography.

We agree that we might have been too approximate in this statement, although we might still agree on the fact that oceanic conditions are typically windy. We propose to withdraw the reference to continental or oceanic climate to make it more general and replace the sentence for the following one: "We conclude that the inference method with a long integration period will lead to very moderate biases for locations with near-neutral conditions and high wind speed, but may lead to much larger bias under stable conditions and low wind speed as soon as the integration period gets up to 12 hours."

There is a strange x on line 468. Font sizes for figure 7 should be increased.

Thanks for spotting the x. It came from a problem when pasting Figure 7. We propose to modify Figure 7 to increase font size and improve as follows:

[Figure]

**Figure 7. Relative root mean squared error as a function of integration period for stability factor and friction velocity classes for a single 25 m side field. Medians and quartiles are given for equally sized bins of $u_*$ and $1/L$ and for the lowest sensor height (0.25 m). The blue, pink and green curves are the 3rd, 2nd and 1st quartiles, respectively.**

741: why bird colonies?

Actually the only references we found where this bias was evaluated were those from emission estimates from bird colonies. We propose to withdraw the mention to bird colonies as this does not add much to the conclusion statement.

742: again, continental does not imply low wind speeds.

As in previous comment we propose to withdraw the reference to continental climate.


This paper presents an appealing and low-cost approach to determine NH3 losses from adjacent multiple agronomical plots by a combination of concentration measurements using passive sampler devices and a dispersion model that is driven by turbulent parameters inferred from standard 30 minutes meteorological data.

The aim of the paper is described as: "Can inverse dispersion modelling approaches be used for inferring NH3 emissions from multiple small plots (agronomic trials) using passive samplers, and to which degree of accuracy?"

The overall answer is encouraging with the statement in the conclusions "In this study we have demonstrated that it is possible to infer with reasonable biases ammonia emissions from multiple small fields located near each other using a combination of a dispersion model and a set of passive diffusion sensors which integrate over a few hours to weekly periods".

According to our judgement the accuracy will be mainly determined by two aspects addressed below.

**a) Bias related to the applied dispersion modelling**

Dispersion models are a simplified mathematical representation of the turbulent motion in the surface layer and will always deviate from reality. Systematic biases can be expected when modelling the lower heights of the measurements that are discussed in this paper. For concentration sensors place close to the ground (e.g. 25cm above ground) transfer functions are likely to be biased due to e.g. the needed simplifications that must be made to describe the exchange process at the ground, the natural heterogeneity on a small scale at the surface or the violation of model assumptions such as the failure of K-theory close to the canopy (Raupach and Legg, 1984). Furthermore, the translation of the sensor height in a model framework is challenging for very low heights since the sensor height value (and with that the resulting value of D at that location) becomes very sensitive to sensor height measurement errors as well as to the absolute values of z0 and d. To us, a sensor height of 25 cm seems too close to the surfaces.

We do agree with A. Neftel and C. Häni concern that model representations may be biased close to the ground, and especially Gaussian like models, since they do not intrinsically account for near-field dispersion as shown by Raupach and Legg (1984) and subsequent publications from M.R. Raupach (Raupach, 1987; Raupach, 1989b). This is not the case of the Langevin models that account for near-field dispersion (Raupach, 1989b; Thomson, 1987). However, as exposed by Raupach (1989a), the height at which the near-field effect is sensible would be equal to $\sigma_w T_L \sim 0.3\ \sigma_w\ h\ /\ u_* \sim 0.3\ x\ 1.25\ x\ h \sim 0.4\ h$ where $\sigma_w$ is the vertical air velocity standard deviation, $h$ is canopy height and $T_L$ is the Lagrangian time scale. Numerical values were derived from Raupach (1989a), Figure 1. Hence we would expect this far field effect to be small on situations with small canopies (or by extension with small roughness height for a bare soil). Typically this would correspond to about 5 cm above a 10 cm canopy and ~1 cm above a 3 cm roughness. Hence we agree with A. Neftel and C. Häni that this would represent a quite important fraction of the sensor height if this sensor would be placed lower than 50 cm (20% for 25 cm). This would especially be critical for canopies that are taller than 10 cm.

Regarding the uncertainty in determining the height of the sensor close to the ground, this is a very sound remark. We however see in Figures 6 and 10 of the manuscript that the method gives similar biases for sensors placed at 25 cm and 50 cm above ground (and also close biases for h = 1 m), for plots of 25 m x 25 m. We hence agree with the concern of A. Neftel and C. Häni that 25 cm would be too low and we should rather target heights of 50 cm. We propose to add this statement in the conclusions: *"Although the lowest sensors have the best condition number, we would rather recommend in practice using heights of 50 cm above the canopy in order to reduce uncertainty in positioning the sensors close to the ground as well as avoid being too close to the roughness layer close to the canopy which is characterised by non-diffusive transfer."*

The authors are using their FIDES-3D model that is based on an analytical solution of the advection-diffusion equation. This model is compared with the backward Lagrangian Stochastic dispersion (bLS) model described in Flesch et al. (2004) (the "WindTrax" software, Thunder Beach Scientific, Nanaimo, Canada). For the presented analysis the FIDES model Kz was adopted to match the far field approximation of Kz of the bLS model. We are missing an explanation, why this was done.

The aim of matching the far-field diffusivity of the two models was to make the FIDES model consistent with the bLS approach, which is a commonly used method nowadays for estimating ammonia emissions with inversion techniques. As exposed in the manuscript, one major difference arises from the fact that the Phillip (1959) approximation of the advection-diffusion equation (which is identical to the approach of Korman and Meixner (2001)) has a Schmidt number equal to 1, while bLS approaches have a Schmidt number equal to 0.64. Wilson (2015) showed that the choice of the Schmidt number has a great effect on footprint, and hence on concentration above a small source and should therefore be explicitly given. Wilson further showed that the difference in footprints predicted by diffusion and Langevin models (like bLS) are small under neutral and stable conditions provided they have similar Schmidt numbers, although the difference in footprints remains large under unstable conditions even with identical Schmidt number (with the Langevin models diffusing less than the Eulerian ones). We hence chose to use an approach that was as close as possible to the bLS approach. To do so, we matched the far field diffusivities of the two models, as this would ensure that the two models would provide similar concentrations at heights larger than a few decimetres. Moreover there was no point in matching near-field dispersion, as FIDES does not account for near-field dispersion.

In the supplement, a detailed investigation is presented how the two models differ in their formulation of the vertical diffusivity Kz. The assumed far field vertical diffusivity in the bLS model is approximated by parametrizations provided in Flesch et al. (1995). We would like to remark that WindTrax uses slightly different default parametrizations of $\sigma_w$ than provided in Flesch et al. (1995) (see e.g. the manual on the WindTrax homepage). This is resulting in vertical diffusivities given as:

$$Kz\,(Z) = 0.5 * 1.25 * u*Z/(1 + 5Z/L) \text{ for z/L} \geq 0 \qquad\qquad \text{(WT1)}$$

$$Kz\,(Z) = 0.5 * 1.25 * u*Z \times (1 - 6Z/L)^{0.25}\,(1 - 3Z/L)^{(1/3)} \text{ for z/L} < 0 \qquad \text{(WT2)}$$

with a Schmidt number value of $Sc \cong 0.64$ for near-neutral stabilities with a smooth transition from L = ∞ to L = -∞. These equations differ from the equations S7 and S8, and imply a different interpretation of the differences between FIDES and WindTrax, though without changing the numeric results of the comparison.

We would like to thank C. Häni and A. Neftel for providing the exact expression used in Windtrax. We indeed only referred to the work of Flesch et al. (1995). These equations WT1 and WT2 are more consistent than those we reported since, as opposed to equations S7 and S8, they insure continuity in $Kz(Z)$ when $z/L \to 0$. We see from the set of equations WT1, WT2, S7 and S8 that $Kz(Z)$ is similar under non-neutral conditions in S7 and WT1 though 4% smaller in Windtrax but that $Kz(Z)$ is 16% smaller in Windtrax (WT2) than in the "tuned FIDES" (S8) under unstable conditions. This therefore explains better Figure S3 which shows a good fit between the "tuned FIDES" and WindTrax under stable and neutral conditions but a lower concentration modelled with the "tuned FIDES" at 2 m above ground under unstable conditions. Indeed, since the far-field diffusivity is larger in the tuned FIDES, this model lead to larger diffusion and hence lower concentrations away from the source. However, another difference comes from constitutive differences between Eulerian and Langevin models under unstable conditions as shown by Wilson (2015).

We have checked that the results reported in this manuscript remain mostly unchanged since already in line with the most important feature of $Sc$ ~ 0.64. We however quantified a difference of around -18% ± 10% in the concentration modelled using equations S7 and S8 (Flesch et al., 1995) compared to equations WT1 and WT2 (Flesch et al., 2004) for a single source of 25 m x 25 m. Since this difference is systematic and since we use the same model for forward and backward modelling we do not expect any impact on the conclusions we have drawn from this study. Indeed, we checked for a single source of 25 m x 25 m that the biased inferred using eq. S7 and S8 and WT1 and WT2 were similar within less than 1% for most cases. Noticeably, the biases for the highest sensors were diminished with WT1 and WT2. We hence propose to leave equations s7 and S8 in section S4.2 as they are but to stipulate explicitly that these correspond to Flesch et al. (1995) and not to WindTrax. We also propose to modify Figure S3 to explicitly mention this (see new Figure 3 below).

In the supplement Figure S3 presents a comparison between evaluated concentrations with FIDES and WindTrax respectively using the prescribed emission sources with the SVAT model. This figure is hiding the apparent differences as a double logarithmic representation is used and the concentrations are shown using an emission source that shows a positive correlation between the meteo input parameters of the models and the source strength. E.g. for neutral conditions the regression of ratio of the concentrations calculated with FIDES and WindTrax at a height of 0.25m is indicated as $cFIDES = 0.97 \cdot cWindtrax^{0.87}$. For a concentration of 1 the ratio is 0.97 and for a concentration of 100 the ratio becomes 0.53. As the transfer function D in FIDES and WindTrax are only depending on the prevailing turbulent parameters it would be more instructive to use a constant unit emission of 1 and show the ratio on a linear scale as function of $u_*$ and $L$ in a similar way as the authors have done in a previous paper (Carozzi et al., 2013).

We thank A. Neftel and C. Häni for this very useful comment. We propose to modify Figure S3 and section S4.3. Below is given the proposed updated Figure S3 showing the comparison between the two models using linear regressions forced to zero and graphs with linear scales. We propose to change the text in the supplementary material to the following: *In Figure 3, we notice that the concentration modelled with the tuned FIDES at 2 m height was lower by roughly 15% compared to WindTrax under unstable conditions but is comparable under stable and neutral conditions. Lower down at 0.25 m height, the tuned FIDES systematically underestimated the concentration by 15-22% whatever the stability.*

[Figure]

**Figure S3. FIDES versus Windtrax concentration modelled above an ammonia source of 25 × 25 m at 0.25 and 2 m heights. The FIDES vertical diffusivity $K_z$ fits the WindTrax $K_z$. The comparison is made over the entire year of 2008 in the FR-Gri ICOS site. S, U and N stand for stable, unstable and neutral atmospheric conditions. The linear regression equation is given for each condition together with the $R^2$ of that regression. The black line is the 1:1 line.**

Below we also computed the graphs showing the concentration residual as a function of 1/L, as suggested by C. Häni and A. Neftel (Figure S4). We propose to include Figure 4 and the following text in the supplementary material: *Figure 4 shows that under unstable conditions FIDES underestimated the concentrations up to 30% at 0.5 m compared to WindTrax, while this gap was reduced and more scattered at 2 m height. Moving towards neutral conditions the two models tend to agree notwithstanding an overestimation of 10% by WindTrax at 2 m height concurrently with an underestimation of the same magnitude by FIDES at 0.5 m. Under stable conditions there was a good agreement at 2 m height, while this agreement remains poorer at lower heights.*

[Figure]

**Figure S4. Relative difference between FIDES and WindTrax concentrations as a function of the stability parameter (1/L). Data refer to the same conditions reported in Figure S3.**

**b) Bias related to the concentration measurements**

The use of passive diffusive samplers is a challenging business. Within different networks the reliability of PS such as ALPHA samplers or Radiellos have been proven, but the use of them close to emitting source showed major deviations compared to other measurements. E.g. Misselbrook et al. (2005) found severe overestimations of passive diffusive samplers. The latest investigation stems from the Dronten experiment and is discussed in a paper by Michael Bell et al. (submitted to AFM). In this experiment the ALPHA samplers were affected by a positive bias in the order of 50% relative to the other devices. We speculated that the exposure of the PS with the protection hat above them cached eddies from below loaded with higher NH3 concentrations but shielded eddies with lower concentrations from above. Figure 1 illustrates the NH3 dynamic that occur over an emitting surface. The concentration was measured with a fast device described in Sintermann et al. (2011). Immediately after application of slurry with a splash plate the NH3 concentration was measured at a height of 1m above ground with an ionization technique and a strongly heated inlet line to avoid as much as possible damping effects.

[Figure]

**Figure 1: NH3 concentration time series measured 1m above ground over a manured surface with splash plate.**

First we notice as A. Neftel and C. Häni that ALPHA badges are very reliable for network measurements concentrations, as for instance showed by the recent Met-NH3 project (http://www.metnh3.eu). We are however very much aware that the use of ALPHA badges close to emitting sources may be biased. The reason for that bias is however unclear and would need comparisons with fast and unbiased sensors. The assessment by Misselbrook et al. (2005) shows that with high concentrations diffusion samplers may lead to overestimation of up to 70% of the concentration. They suggest potential issues related to the deformation of the Teflon membrane which would modify the distance between coated filters and the Teflon membrane that could cause sampler saturation.

The speculations from A. Neftel and C. Häni are interesting. One could indeed speculate that sweeps, which dominate turbulent transport near the top of the canopy and are characterised by lower wind speed with positive vertical velocity (Poggi and Katul, 2007), could lead to an artificial build-up of the concentration underneath the protecting caps. We could speculate that ejections would not be efficient in "purging" the volume underneath the cap and hence letting over time the concentration being higher in this area. One could also speculate on adsorption-desorption of ammonia on the walls of the ALPHA badges that would be non-linear in response to $NH_3$ concentrations and lead to possible over estimations under highly fluctuating concentrations as shown in Figure 1 above. This issue necessitates experimental validation of the methodology anyway. We hence propose to add the following text in the discussions section 3.6: *Misselbrook et al. (2005) compared different methodologies and showed that under high concentrations diffusion samplers may lead to overestimation of up to 70% of the concentration. They suggest potential issues related to the deformation of the Teflon membrane which would modify the distance between coated filters and the Teflon membrane that could cause sampler saturation. There is hence some concern on the quality of diffusion samplers to measure concentrations close to large sources which would necessitate field validations.*

We also propose to add a sentence at the end of the conclusion to emphasize this issue for future work: *"Special care should be taken in validating the use of ALPHA samplers near very strong sources"*

**Concluding comments:**

We judge that the most important potential biases of the proposed multiplot approach are related to biases of the concentration measurements and the used dispersion coefficient. It would be instructive to calculate probability density functions of the estimated emissions with a dataset that reflect the distributions of the measurements and the turbulence parameters that drive the dispersion model.

This would indeed be very instructive but we feel that this issue is rather a work to be for a next study. Indeed, in this study we have explored the first order variability which is driven by the change in meteorological conditions observed in the 13 periods over a year in typical western European climate. The next step could be to extend this assessment to other datasets part of Fluxnet network to incorporate more continental climate conditions. The script we developed for this study actually incorporates measurement noise but we disabled this feature for calculation time reasons.

The authors have tested their setup in field trial in April 2011 applying slurry with a DM content of 6% and an application rate of 41 kg N-NH3/ha. According to the details given in the text, we assume that broadband application was used and was compared to fast incorporation and no application. The cumulated loss amounted to 8 to 10% of the applied NH3. For broadband application, this is a loss on the low side (see e.g. Häni et al.,2016). We would not be astonished if the real emissions would be double as high.

We would like to thank very much A. Neftel and C. Häni for their question and to grant them for their guess. Indeed, we double checked the calculation script used for the real test case and we found one bug in the calculation of the cumulated NH$_3$ emissions: the multiplicative constant to account for the number of second per time step was set to 30 min while the time step of that particular test case was 60 min. Since a 60 min time step is unusual we did not spot this error in the first place, but we have since then changed the script to calculate the time step from the meteorological dataset. We used this new script and found this bug. This change results in a doubled emission compared to what was given in the discussion manuscript. We hence find, as guessed by A. Neftel and C. Häni that the cumulated losses represented around 20% of the nitrogen applied. We propose to change Figures 16 and 17 as below:

[Figure]

**Figure 16. Cumulated fluxes estimated with the inference method on the real test case with strategy C7. Three treatments with two repetitions are compared.**

[Figure]

**Figure 17.** Same as Figure 16 but grouped by treatments and with additional strategies C4 and C6 which consider that replicates have the same surface flux. The variability in the boxplot aggregates the uncertainty on the inference method (the standard deviation on the flux estimate in the least-square model, which accounts for the variability in the replicated concentration measurements), and the variability between the repetitions in each treatment. Letters *a, b* and *c* show significant differences between treatments for the C7 strategy, according to a Tukey test (95% family-wise confidence level).

We also propose to modify the text in section 3.5 (lines 630-636) to: *Surface slurry application showed the largest emissions: $9 \pm 0.3$ kg N ha$^{-1}$ in B1 and $10 \pm 0.2$ kg N ha$^{-1}$ in B2 (median and confidence interval). This corresponds to an emission factor around 23% of the N-NH$_4$ applied and 8% of the total N applied, which is in-line with agronomic references (Sintermann et al., 2011a; Sommer et al., 2006). In contrast, the incorporated slurry showed much smaller emissions: $0.3 \pm 0.2$ kg N ha$^{-1}$ in B1 and $0.6 \pm 0.2$ kg N ha$^{-1}$ in B2. It is noticeable that the no-application showed slight deposition, especially in B2: $-0.26 \pm 0.2$ kg N ha$^{-1}$ in B1 and $-1.7 \pm 0.2$ kg N ha$^{-1}$ in B2.*

We also propose to change lines 659-663 as follows: *Therefore, we could expect that the real flux is the one measured with C7 times 1.15 ($\pm$ 0.08), hence would be $10.9 \pm 1.3$ kg N ha$^{-1}$. This corresponds to $27 \pm 3$ % of the N-NH$_4$ applied and ~$9 \pm 1$% of the total N applied. For the incorporated slurry, the emissions are around 20 times smaller than the emissions from the surface applied slurry. Under these conditions, the bias on the emission would be around -20%, which means that the corrected emissions would range from 0.5% to 2.5% of the N-NH$_4$ applied and 0.2 and 0.8% of the total N applied.*

The presented approach to perform NH3 emission measurements in a multiplot arrangement is encouraging and goes in a good direction. To make the approach more robust, the employed ALPHA NH3 sampling systems should be validated under real conditions, i.e. over an emitting source in comparison with e.g. MiniDOAS systems (Sintermann et al., 2016).

We completely agree with A. Neftel and C. Häni about this issue. Although the ALPHA badges have shown to be very precise in comparisons under laboratory and field conditions, it is worth comparing them with an independent technique in a situation where the source is small and intense and where the sensor is placed near the ground (e.g. 50 cm above ground).

Finally, we would like to invite the authors to collaborate with us to compare the FIDES and WindTrax approach. We have an extensive dataset from field trials where we released CH4 or a mixture of NH3 and CH4 from a circular artificial source with a diameter of 20 meters (Häni et al., 2017).

We are honoured by this invitation to collaborate and would be very happy to compare our approach with WindTrax on CH4 and NH3 emissions.